# Co-option of neurotransmitter signaling for inter-organismal communication in *C. elegans*

Christopher D. Chute[1,5], Elizabeth M. DiLoreto[1], Ying K. Zhang[2], Douglas K. Reilly[1], Diego Rayes [3,6], Veronica L. Coyle[1,7], Hee June Choi[4], Mark J. Alkema[3], Frank C. Schroeder [2] & Jagan Srinivasan [1]

Biogenic amine neurotransmitters play a central role in metazoan biology, and both their chemical structures and cognate receptors are evolutionarily conserved. Their primary roles are in cell-to-cell signaling, as biogenic amines are not normally recruited for communication between separate individuals. Here, we show that in the nematode *C. elegans*, a neurotransmitter-sensing G protein-coupled receptor, TYRA-2, is required for avoidance responses to osas#9, an ascaroside pheromone that incorporates the neurotransmitter, octopamine. Neuronal ablation, cell-specific genetic rescue, and calcium imaging show that *tyra-2* expression in the nociceptive neuron, ASH, is necessary and sufficient to induce osas#9 avoidance. Ectopic expression in the AWA neuron, which is generally associated with attractive responses, reverses the response to osas#9, resulting in attraction instead of avoidance behavior, confirming that TYRA-2 partakes in the sensing of osas#9. The TYRA-2/osas#9 signaling system represents an inter-organismal communication channel that evolved via co-option of a neurotransmitter and its cognate receptor.

[1] Biology and Biotechnology Department, Worcester Polytechnic Institute, Worcester, MA 01605, USA. [2] Boyce Thompson Institute and Department of Chemistry and Chemical Biology, Cornell University, Ithaca, NY 14853, USA. [3] Neurobiology Department, University of Massachusetts Medical School, Worcester, MA 01605, USA. [4] Biomedical Engineering Department, Worcester Polytechnic Institute, Worcester, MA 01605, USA. [5] Present address: BioHelix Corporation, Beverly, MA 01915, USA. [6] Present address: Instituto de Investigaciones Bioquímicas de Bahía Blanca (CONICET), Departamento de Biología, Bioquímica y Farmacia, Universidad Nacional del Sur, Bahía Blanca B8000, Argentina. [7] Present address: AbbVie, Cambridge, MA 02139, USA. Correspondence and requests for materials should be addressed to J.S. (email: jsrinivasan@wpi.edu)

Inter-organismal communication occurs in many forms across the animal kingdom, both within and between species[1]. Chemosensation, both ancient and ubiquitous across all kingdoms of life, underlies social responses mediated by chemical communication[2]. Social chemical communication requires both cell-to-cell and inter- organismal signaling. First, a chemical cue is released into the environment by one organism that is then detected by specific receptors in another organism. Upon sensation, inter-cellular signaling pathways, e.g., neurotransmitter signaling, are activated that ultimately coordinate a social response.

Neurotransmitter monoamines, such as dopamine, serotonin, tyramine, and octopamine, serve diverse functions across kingdoms[3]. The associated signaling pathways often rely on highly regulated biosyntheses, translocation (either by way of diffusion or through active transport), and perception by dedicated chemoreceptors. Many neurotransmitters are perceived via G protein-coupled receptors (GPCRs); in fact, there is a close relationship between GPCR diversification and neurotransmitter synthesis in shaping neuronal systems[4]. Notably, the most common neurotransmitters share similar behavioral functions across phyla. For example, serotonin is commonly involved in regulating food responses[5]. Other neurotransmitters, such as tyramine and octopamine, are only found in trace amounts in vertebrates, and act as adrenergic signaling compound in invertebrates[6].

The nematode *Caenorhabditis elegans* offers many advantages for studying social chemical communication and neuronal signaling, namely the animal's tractability, well-characterized nervous system, and robust social behavioral responses to pheromones[7]. *C. elegans* secretes a class of small molecules, the ascaroside pheromones, which serve diverse functions in inter-organismal chemical signaling[8,9]. As a core feature, these molecules include an ascarylose sugar attached to a fatty acid-derived side chain that can be optionally decorated with building blocks from other primary metabolic pathways[9]. Ascaroside production, and thus the profile of relayed chemical messages, is strongly dependent on the animal's sex, life stage, environment, and physiological state[10–12]. Depending on their specific chemical structures and concentration, the effects of ascaroside signaling vary from social (e.g. attraction to icas#3) to developmental (e.g. induction of dauer by ascr#8; Fig. 1a)[12–15]. Furthermore, different combinations of these ascarosides can act synergistically to elicit a stronger behavioral response than one ascaroside alone, such as male attraction to ascr#2, ascr#3, and ascr#4[13]. Several GPCRs have been identified as chemoreceptors of ascaroside pheromones, such as SRX-43 in ASI to promote dwelling behavior, and DAF-37 in ASK regulating hermaphrodite repulsion[16–19].

Recently, an ascaroside, named osas#9, that incorporates the neurotransmitter octopamine, was identified[10]. osas#9 is produced in large quantities specifically by starved L1 larvae, and elicits aversive responses in starved, but not well fed conspecifics[10]. The dependency on starvation of both its production and elicited response suggests osas#9 relays information on physiological status and unfavorable foraging conditions. However, it is unknown how osas#9 is perceived and drives starvation-dependent behavioral responses. Based on the unusual incorporation of a monoamine neurotransmitter building block in osas#9, we asked whether other components of monoamine signaling pathways have been recruited for inter-organismal signaling via osas#9. Here, we show that TYRA-2, an endogenous trace amine receptor, is required for the perception of osas#9, demonstrating co-option of a neurotransmitter and a neurotransmitter receptor for inter-organismal communication.

## Results

**Aversive responses to osas#9 require the GPCR TYRA-2.** Previous work has shown that production of the ascaroside osas#9 (Fig. 1a) is starkly increased in starved L1 larvae, and elicits avoidance behavior in starved young adult hermaphrodites (Fig. 1b)[10]. This starvation-dependent response is reversible: when worms are starved for an hour, and then reintroduced to food for two hours, no avoidance behavior is observed (Supplementary Fig. 1A). In this study, we tested a broader range of conditions, and found that osas#9 elicits avoidance regardless of the sex or the developmental stage of the animal (Fig. 1c), and that osas#9 is active over a broad range of concentrations (fM - μM; Supplementary Fig. 1B). 1 μM osas#9 was used for the remainder of this study unless otherwise noted (Fig. 1d). Ascarosides, such as the male attractant, ascr#3, and aggregation signal, icas#3, show activity profiles that are as similarly broad as that of osas#9, whereas others, such as the mating cue, ascr#8, are active only within narrow concentration ranges[13,20,21].

The chemical structure of osas#9 is unusual in that it includes the neurotransmitter octopamine as a building block (Fig. 1a). Because both octopamine, and the biosynthetically related tyramine, play important roles in orchestrating starvation responses, we investigated receptors of octopamine (*ser-3, ser-6, and octr-1*) and tyramine (*tyra-2, tyra-3, ser-2, and ser-3*) for involvement in the osas#9 response (Fig. 2a)[22–26]. We found that avoidance to osas#9 is largely abolished in a *tyra-2* loss of function (*lof*) mutant, whereas osas#9 avoidance was largely unaffected in the other neurotransmitter receptor mutants (Fig. 2a). We confirmed this phenotype was a result of the loss of *tyra-2* by testing a second *tyra-2 lof* allele (Fig. 2b), and by neuron-targeted knockdown of *tyra-2* (Supplementary Fig. 2A, B)[27–29].

TYRA-2 is a G protein-coupled receptor (GPCR) that has been shown to bind tyramine with high affinity and, to a lesser extent, octopamine[24]. To exclude the possibility that *tyra-2* is necessary for general avoidance behaviors, we subjected *tyra-2 lof* animals to three well-studied chemical deterrents: sodium dodecyl sulfate (SDS), copper chloride ($CuCl_2$), and glycerol. No defects were found in *tyra-2 lof* animals' ability to respond aversively to these deterrents (Fig. 2c). This indicates that *tyra-2* is specifically required for osas#9 avoidance, and is not part of a generalized unisensory avoidance response circuit. To determine the presence of receptors other than TYRA-2 that contribute to the sensation of osas#9, we exposed wild type and *tyra-2 lof* animals to increasing concentrations of osas#9 (Fig. 2d). Wild-type worms avoided osas#9 at all concentrations tested, whereas neither *tyra-2 lof* mutant avoided osas#9 at 1 μM or 10 μM concentrations (Fig. 2d). However, at 100 μM, both *tyra-2 lof* strains exhibited robust avoidance to osas#9, suggesting that other receptors respond to osas#9 at non-physiological concentrations (Fig. 2d).

Since the response to osas#9 is dependent on physiological state, we examined whether *tyra-2* transcript levels changed under starvation conditions using quantitative RT-qPCR. Starved worms exhibited a nearly two-fold increase in *tyra-2* expression (Fig. 2e).

Next we asked whether tyramine signaling is required for the osas#9 avoidance response, as *tyra-2* is known to bind to endogenous tyramine[24]. We assayed two *tdc-1 lof* mutants, which lack the ability to synthesize tyramine[30]. We observed that the behavioral response to osas#9 was unaltered in animals lacking tyramine biosynthesis (Fig. 2f), demonstrating that the role of TYRA-2 in osas#9 avoidance is independent of tyramine, suggesting that TYRA-2 may be involved in perception of a ligand other than tyramine in promoting the aversive response to osas#9.

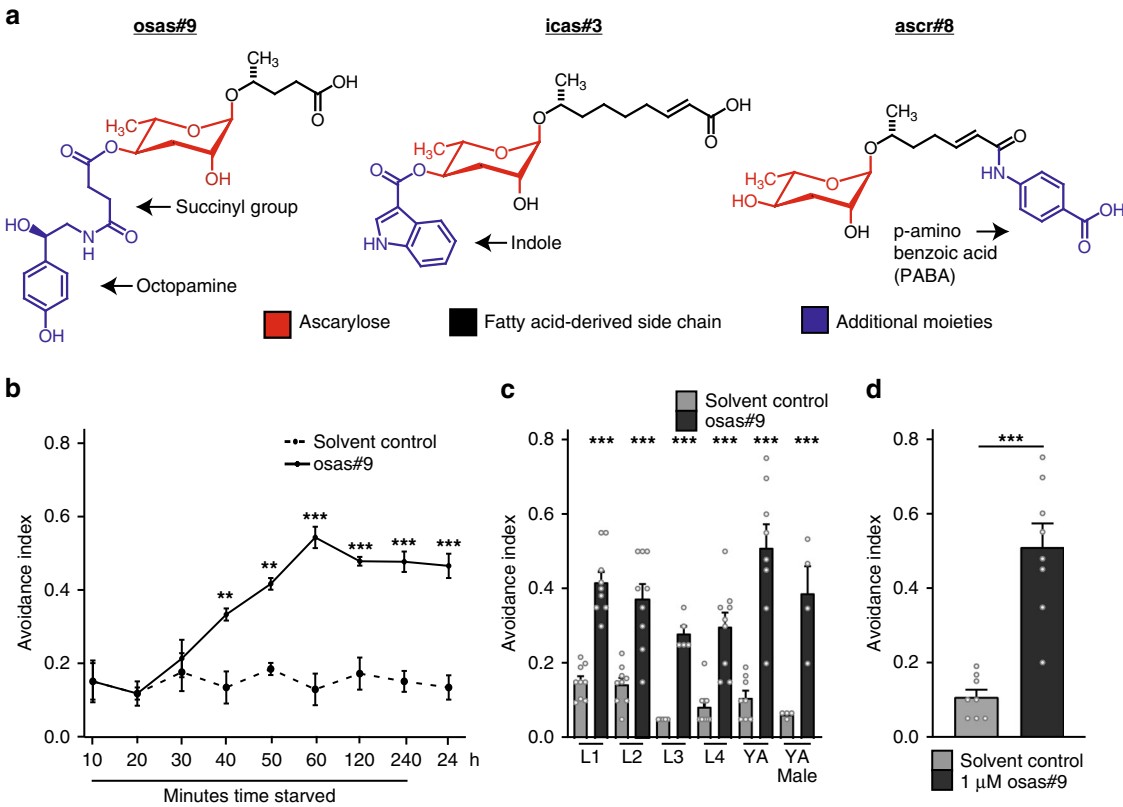

**Fig. 1** osas#9 is repulsive to starved animals. **a** Structural and functional diversity of ascarosides. osas#9 is involved in avoidance, icas#3 attracts hermaphrodites, and ascr#8 attracts males at low concentrations and induces dauer formation at high concentrations. **b** Avoidance to osas#9 is dependent on the physiological state of *C. elegans*. Avoidance index of young adult (YA), wild type (N2) animals in response to solvent control (black dotted line), and 1 μM osas#9 (black solid line) after different time points after removal from food. After 40 minutes of starvation, animals begin to avoid osas#9, with the response reaching a plateau at 60 min. $n \geq 3$ trials. Note for all other assays, unless otherwise stated, animals are starved for at least 60 min. **c** All life stages of hermaphrodites, and adult males, avoid osas#9 when starved. $n \geq 4$ trials. **d** Avoidance index for starved young adult (YA) wild-type worms in response 1 μM osas#9. $n = 8$ trials. 1 μM osas#9 concentration was used in all other assays unless stated otherwise. Data presented as mean ± S.E.M; $*p < 0.05$, $**p < 0.01$, $***p < 0.001$, one factor ANOVA with Sidak's multiple comparison posttest, except for Fig. 1d, where a Student's *t*-test was used. Individual data points for each bar graph are represented as gray circles

***tyra-2 is required in the ASH neurons for osas#9 sensation***. To better understand its role in the osas#9 aversion pathway we then asked where *tyra-2* is expressed and localized. For this purpose, we designed a *tyra-2* translational fusion construct consisting of the entire genomic locus, including 2 kb upstream, fused to GFP (p*tyra-2::tyra-2::GFP*). We observed TYRA-2 expression in four sensory neurons: ASH, ASE, ASG, and ASI (Fig. 3a); as well as the pharyngeal motor neuron, NSM. These results are in agreement with previous *tyra-2* expression studies[24] (Fig. 3a). We laser-ablated individual amphid sensory neurons to determine if a *tyra-2* expressing sensory neuron is required for the response. This revealed that ASH neurons are required for osas#9 response, whereas ablation of other neurons did not have a strong effect (Fig. 3b). We observed a slight reduction in the magnitude of the osas#9 aversive response in ASE and ASI laser-ablated animals (Fig. 3b), although ASH/ASE and ASH/ASI double ablated animals did not differ in response from animals with ASH ablated alone, and ASE/ASI ablated animals did not differ from ASE or ASI alone (Fig. 3b). To determine whether other sensory neurons played a role in mediating osas#9 avoidance, we tested genetic ablation lines of ASH, ASE, and ASI neurons[31–34]. We observed that at all tested concentrations, only the ASH genetic ablation line showed complete abolishment of osas#9 avoidance (Supplementary Fig. 3A–C). As with the laser ablation studies, we observed a slight decrease in osas#9 avoidance in ASE and ASI ablated animals (Supplementary Fig. 3A–C). Ablation of neurons

not expressing *tyra-2* did not result in any defects in the response to osas#9 (Supplementary Fig. 3D). Our data implies that osas#9 is primarily sensed by the ASH sensory neurons, while the ASE and ASI sensory neurons may contribute to the sensation, possibly by sensitizing ASH sensory neurons or by regulating downstream interneurons within the osas#9 response circuitry[35].

To further elucidate the role of the ASH sensory neurons and TYRA-2 in osas#9 sensation, we utilized a microfluidic olfactory imaging chip that enables detection of calcium transients in sensory neurons[36,37]. We observed that upon exposure to 1 μM osas#9, wild type animals expressing GCaMP3 in the ASH neurons exhibit robust increase in fluorescence (Fig. 3c, d and Supplementary Movie 1). Worms lacking *tyra-2* displayed no fluorescence change upon osas#9 exposure (Fig. 3c, d). Exposure of worms to different concentrations of osas#9 elicited calcium changes in the ASH sensory neuron, correlating with avoidance responses observed at those concentrations (Fig. 3e, f, Supplementary Fig. 1B).

To test whether the observed ASH calcium signals are the result of direct sensation of osas#9 or are induced indirectly in response to osas#9 sensation in other neurons, such as ASE or ASI (Fig. 3b), we used genetic mutants that disrupted either synaptic signaling (*unc-13*)[38] or peptidergic signaling (*unc-31*)[39] (Supplementary Fig. 3E, F). In both *unc-13* and *unc-31* mutants, ASH neurons still displayed an increase in calcium levels upon osas#9 stimulation, indicating that ASH neurons can respond to

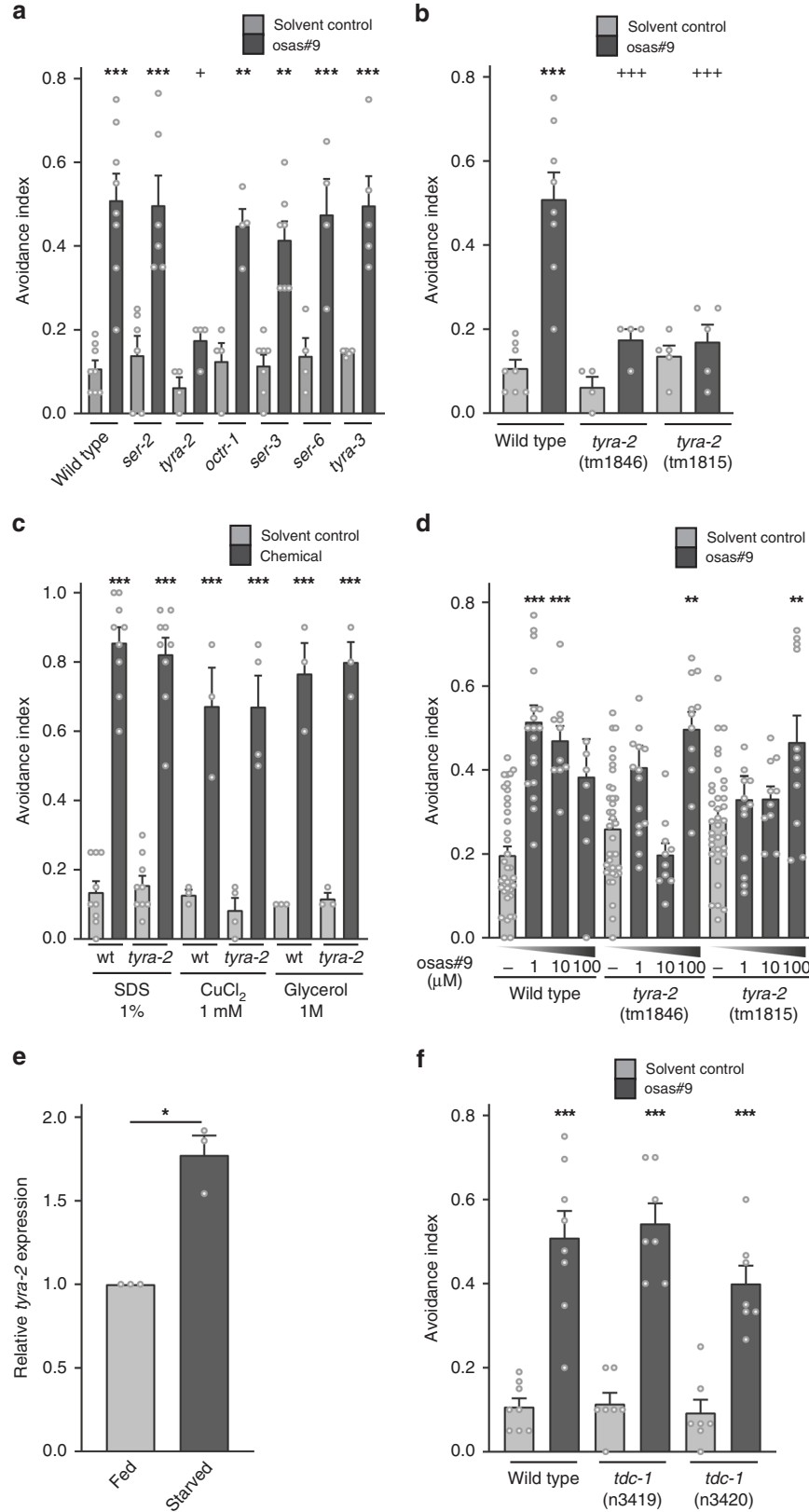

osas#9 independent of feedback signals. However, the magnitude of osas#9-evoked ASH responses was significantly reduced in both *unc-13* and *unc-31* mutants. These data indicate that synaptic and peptidergic mechanisms contribute to the magnitude of the ASH response, suggesting that a distributed circuit modulates osas#9 dynamics, potentially involving other neurons

such as ASE and ASI, as observed for other chemical stimuli in *C. elegans*[35,40].

To test whether ASH neurons can detect other small molecules, we used an unrelated ascaroside, ascr#3, which elicits male attraction in *C. elegans*[13]. Stimulating ASH neurons with ascr#3 did not result in any calcium transients (Supplementary Fig. 3G,

**Fig. 2** *tyra-2* is required for osas#9 mediated aversive responses independent of tyramine. **a** Screen for receptors required to mediate osas#9 avoidance. *tyra-2 lof* animals are defective in osas#9 avoidance response. $n \geq 4$ trials. **b** Two alleles of *tyra-2 lof* animals (tm1846 and tm1815), are defective in osas#9 avoidance behavior. $n \geq 4$ trials. **c** wild type, wt, and *tyra-2 lof* mutants showed no significant differences when subjected to known chemical deterrents. $n \geq 3$ trials. **d** Wild type animals avoided osas#9 at 1 and 10 μM, while both *tyra-2 lof* mutants avoid at only 100 μM. $n \geq 8$ trials. **e** Expression of *tyra-2* receptor is dependent on the physiological state of the animal. RT-qPCR analysis starved animals indicates a nearly twofold upregulation of *tyra-2*. Data are displayed as the ratio of endogenous *tyra-2* mRNA to *ama-1* mRNA from three independent RT-qPCR experiments. $n = 3$ trials. **f** osas#9 avoidance response is not dependent on endogenous tyramine. Two different alleles of *tdc-1 lof* animals (n3419 and n3420) which are deficient in tyramine biosynthesis, exhibit normal response to osas#9. $n \geq 7$ trials. Data presented as mean ± S.E.M; *$p < 0.05$, **$p < 0.01$, ***$p < 0.001$, one factor ANOVA with Sidak's multiple comparison posttest except for 2D and 2E, where Student's *t*-test was used

---

H). Given that tyramine and octopamine are known ligands of TYRA-2, we also tested whether these neurotransmitters elicit aversive responses in *C. elegans*[24]. Both biogenic amines exhibited aversive behaviors at non-physiological concentrations much higher than required for osas#9, i.e., 1 mM for tyramine and octopamine compared to 1 μM for osas#9 (Fig. 3g, Supplementary Fig. 4A). Similarly, this high concentration of tyramine was required to elicited calcium transients in ASH::GCaMP3 line, whereas lower concentrations (1 μM) did not (Fig. 3h, i). Worms exposed to 1 mM octopamine also only displayed minimal changes in calcium transients (Supplementary Fig. 4B, C).

Previous studies have shown that both tyramine and octopamine inhibit serotonin-mediated food-dependent aversive responses to dilute octanol via specific GPCRs[26]. To test whether tyramine inhibits osas#9 sensation, we performed a competition experiment where we exposed worms to different ratios of tyramine and osas#9 (Supplementary Fig. 4D). Wild type worms exposed to equimolar concentrations of tyramine and osas#9 displayed robust avoidance mediated by the TYRA-2 receptor (Supplementary Fig. 4D). Mixtures containing very high (1 mM) concentrations of tyramine elicited aversion in both wild type and *tyra-2 lof* worms, as in the experiments with pure tyramine (Supplementary Fig. 4D). These results, indicate that expression of the TYRA-2 receptor in the ASH sensory neurons is specifically involved in the response to osas#9.

**tyra-2 expression confers the ability to sense osas#9.** Since expression of *tyra-2* in the ASH sensory neurons is required for osas#9 elicited calcium transients, we asked whether *tyra-2* expression in the ASH neurons is sufficient to rescue the osas#9 behavioral response in *tyra-2 lof* animals. Expression of *tyra-2* under the *nhr-79* promoter, which resulted in expression in the ASH and ADL neurons, fully restored osas#9 avoidance (Fig. 4a, b)[41]. We then tested whether rescue of *tyra-2* expression rescues the neurophysiological properties in ASH neurons upon osas#9 exposure. We generated a line expressing GCaMP3 in the ASH neuron in the transgenic rescue line and observed that osas#9 exposure elicited calcium transients similar to wild type animals (Fig. 4c, d).

To demonstrate sub-cellular localization of the TYRA-2 protein in the sensory neurons, we injected the translational reporter generated in Fig. 3a at a lower concentration (1 ng/μl) into *tyra-2 lof* animals, as this has been found to improve receptor localization in the sensory cilia (Maurya and Sengupta, personal communication). The worms expressing the transgene displayed sub-cellular localization in the ASH sensory cilia (Fig. 4e). osas#9 aversion is rescued in these animals, indicating that the transgene is functional (Fig. 4f). These results affirm that the aversive behavioral response to osas#9 is dependent on TYRA-2 localization in the ASH neuronal cilia.

Previous studies in *C. elegans* indicate that behavioral responses (such as aversion or attraction) elicited by an odorant are specified by the olfactory neuron in which the receptor is present, rather than by the olfactory receptor itself[17,42]. We asked

whether driving TYRA-2 receptor expression in other sensory neurons will drive behavioral response to osas#9. For this purpose, we ablated the ASH neurons in the p*nhr-79::tyra-2* strain, in which *tyra-2* is expressed in the ASH and ADL neurons (Fig. 4a, b). ADL neurons have also been shown to detect aversive stimuli[43]. We found that these ASH-ablated transgenic animals still avoid osas#9, similar to ADL ablated worms from this rescue line (Fig. 5a, b). Ablation of both the ASH and ADL neurons in this strain abolished the avoidance response (Fig. 5a, b). This implies that misexpression of *tyra-2* in the ADL neurons is sufficient to confer avoidance to osas#9.

We then asked whether expression of TYRA-2 in AWA neurons, which are generally involved in attractive responses to chemical cues[44], would switch the behavioral valence of osas#9, resulting in attraction to osas#9, instead of aversion. Misexpression of *tyra-2* in the AWA sensory neurons in a *tyra-2 lof* background did not result in avoidance of osas#9, in contrast to expression of *tyra-2* in the ASH neurons (Fig. 5c). We tested wild-type worms and transgenic lines expressing *tyra-2* in the AWA neurons for diacetyl chemotaxis. We observed no discernable difference in their ability to chemotax to the chemical, nor was there any observed defect in speed (Supplementary Fig. 5A, B). In addition, we measured calcium transients in these transgenic worms and found that they respond normally to diacetyl like wild type worms (Supplementary Fig. 5C, D). These results confirmed that ectopic expression of *tyra-2* in AWA sensory neurons did not alter the native chemosensory parameters of AWA neurons. However, upon exposure to 1 μM osas#9, these transgenic worms exhibited negative calcium transients in AWA neurons (Fig. 5d, e), suggesting that AWA::*tyra-2* can sense the chemical. We then used a simple leaving assay to test whether worms expressing *tyra-2* in the AWA neurons display attractive behavior. This assay involves the placement of animals into the center of a NGM agar plate where osas#9 is present, and measuring the distance of animals from the origin in 1-min intervals (Fig. 5f). *tyra-2 lof* animals displayed osas#9 leaving rates equal to the solvent control (Fig. 5g, Supplementary Fig. 6), whereas AWA::*tyra-2* worms displayed osas#9 leaving rates lower than that for solvent controls, indicating attraction-like behavior (Fig. 5g, Supplementary Fig. 6). Furthermore, worms misexpressing *tyra-2* in the AWA neurons stayed significantly closer to the origin than either wild type or *tyra-2 lof* animals when exposed to osas#9 (Fig. 5g, Supplementary Fig. 6). Previous studies have shown that diacetyl stimulation results in increased calcium transients in AWA neurons resulting in suppressed turning behavior[45]. Given this observation, we asked whether reduction of calcium changes in AWA neurons upon osas#9 stimulation will result in increased reversals in AWA::*tyra-2* worms. We observed that AWA::*tyra-2* animals reverse roughly two times as often as wild type or *tyra-2 lof* animals when exposed to osas#9 (Fig. 5h). Hence, misexpression of *tyra-2* in AWA neurons resulted in reprogramming of the worms' behavioral circuits, promoting attraction to the normally aversive compound osas#9.

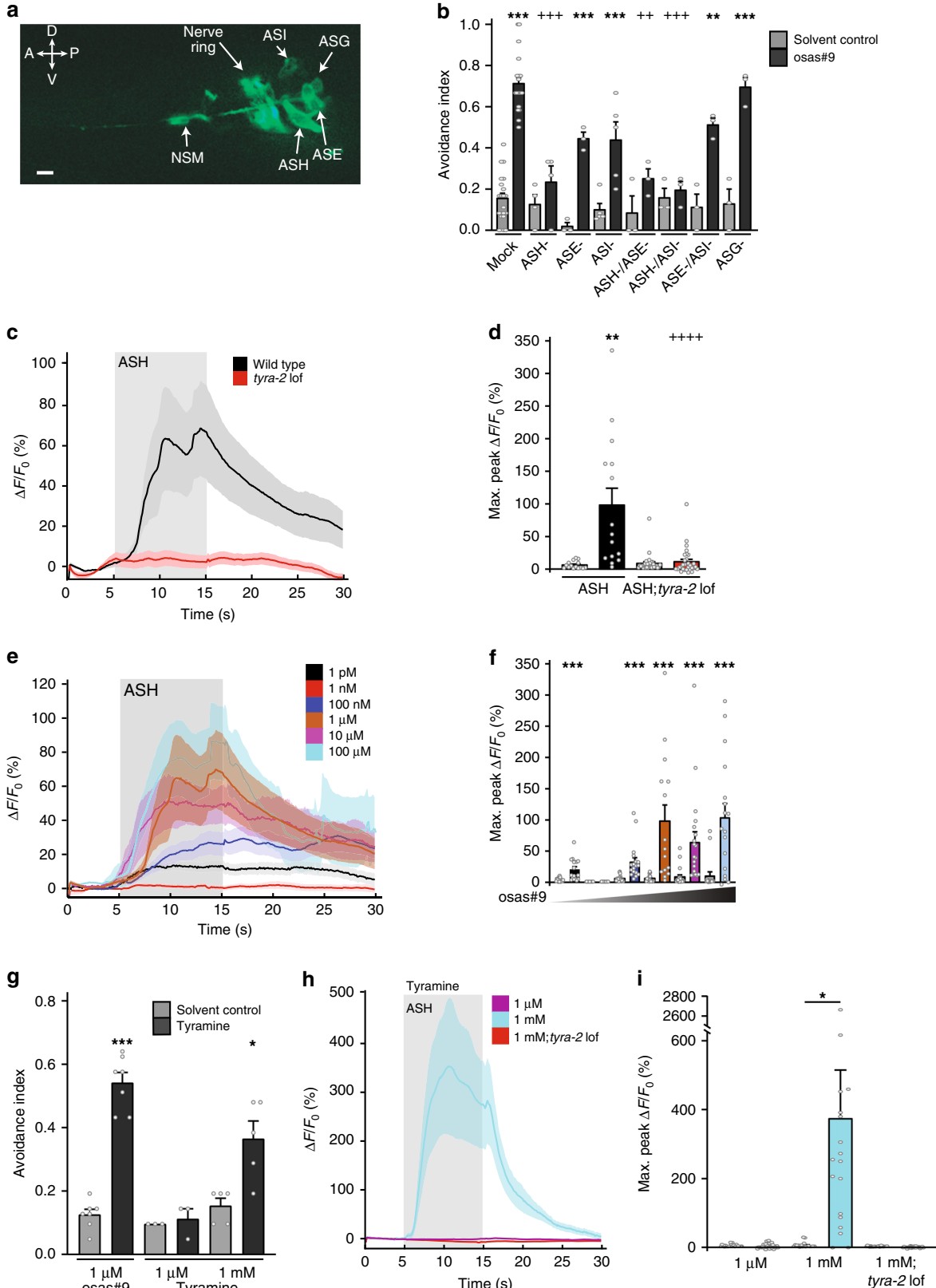

**gpa-6 is necessary in ASH for osas#9 avoidance.** Since expression of the GPCR, TYRA-2, in ASH neurons is required for the osas#9 response, we sought to identify the Gα subunit necessary for osas#9 avoidance. Eight of the twenty-one Gα proteins are expressed in subsets of neurons that include the ASH sensory pair (*gpa-1*, *gpa-3*, *gpa-6*, *gpa-11*, *gpa-13*, *gpa-14*, *gpa-15*, and

*odr-3*)[46,47]. We tested mutants for each of those eight Gα subunits for their response to osas#9 (Fig. 6a), and found that *gpa-6 lof* animals do not avoid osas#9 (Fig. 6a). To determine whether *gpa-6* is necessary in ASH sensory neurons to mediate osas#9 responses, we expressed *gpa-6* using p*nhr-79* in the ASH neurons in a *gpa-6 lof* background. These animals displayed osas#9

**Fig. 3** TYRA-2 is required in ASH for sensation of osas#9. **a** Cellular localization of TYRA-2 in sensory neurons. Expression is seen in ASE, ASG, ASH, ASH, and NSM neurons (×40 magnification, scale bar denotes 10 μm). **b** Chemosensory neurons required for osas#9 response. Neurons expressing *tyra-2* reporter were ablated using laser microbeam. ASH ablations resulted in abolished response to osas#9 that was indistinguishable from solvent control. ASE and ASI ablated animals showed reduced but not complete loss of avoidance. $n \geq 3$ trials ($\geq 10$ ablated animals precondition). **c, d** Calcium dynamics of ASH upon osas#9 exposure. **c** ASH::GCaMP3 animals (black) exhibit calcium transients when exposed to 1 μM osas#9. *tyra-2 lof* animals (red) did not display a change in fluorescence upon stimulation. Shaded region gray depicts time animals were subjected to osas#9, $n = 10$ animals. **d** Maximum fluorescence intensity before (solvent control, gray) and during exposure to 1 μM osas#9 of ASH::GCaMP3 animals (black) and *tyra-2 lof* animals (red). **e** osas#9 elicits calcium transients in ASH neurons at a broad range of concentrations: 1 pM (black), 1 nM (red), 100 nM (blue), 1 μM (orange), 10 μM (magenta), 100 μM (cyan), $n \geq 10$ animals per condition. **f** Maximum fluorescence intensity before and during exposure to varying osas#9 concentrations. **g** Tyramine elicits avoidance only at high concentrations in wild-type animals, $n \geq 5$ trials. **h** Calcium dynamics in ASH upon exposure to different concentrations of tyramine, for ASH::GCaMP3 1 μM (magenta) and 1 mM (cyan) and *tyra-2 lof* 1 mM tyramine (red). Tyramine exposure resulted in a significant increase in calcium transients in ASH at concentrations of 1 mM, but not 1 μM, $n \geq 10$ animals. **i** Maximum fluorescence intensity before (solvent control) and during exposure to varying tyramine concentrations for ASH::GCaMP3 animals and *tyra-2 lof* animals. Data presented as mean ± S.E.M; \*$p < 0.05$, \*\*$p < 0.01$, \*\*\*$p < 0.001$. Figure 3b, one factor ANOVA with Sidak's multiple comparison posttest. Figure 3d, f, i Student's *t*-test was used to compare the solvent control to stimulus max peak fluorescence

avoidance behavior similar to wild type worms (Fig. 6b). To characterize cellular and sub-cellular localization of the *gpa-6* Gα subunit, we created a full-length RFP translational fusion of the entire *gpa-6* locus including 4 kb of the upstream sequence. We observed *gpa-6* expression in the soma of AWA and ASH sensory neurons (Fig. 6c), in agreement with previous studies[46]. However in addition to ASH soma localization, the translational fusion revealed presence of *gpa-6* in ASH cilia (Fig. 6c). Behavioral rescue by *gpa-6* expression specifically in the ASH neurons and its ciliary localization, support that this Gα subunit functions in mediating osas#9 avoidance.

## Discussion

How does a worm survive in changing environmental and physiological conditions? Given *C. elegans*' complex ecology and boom-and-bust lifestyle, worms need to make frequent adaptive developmental and physiological choices[48]. The octopamine-derived pheromone, osas#9, secreted in large quantities by L1 larvae under starvation conditions, appears to promote dispersal away from unfavorable conditions (Fig. 7). Here we show that this pheromone is detected by the GPCR, TYRA-2, a canonical neurotransmitter receptor expressed in the ASH sensory neurons. To our knowledge, this is the first instance in which a "repurposed internal receptor" partakes in pheromone perception. Notably, octopamine, the distinguishing structural feature of osas#9, has been implicated in responses to food scarcity in invertebrates, including insects[49–51], *C. elegans*[22,52–55], and molluscs[56]. Our findings indicate that worms navigate adverse environmental conditions, in part, using social communication networks that employ signaling molecules and receptors derived from relevant endocrine signaling pathways (Fig. 7).

TYRA-2 has previously been shown to contain the conserved Asp$^{3.32}$ residue required for amine binding, allowing the receptor to bind tyramine with high affinity and, to a lesser extent, octopamine[24]. In contrast, osas#9 lacks a basic amine, instead containing an amide, as well as an acidic sidechain. These chemical considerations suggest that TYRA-2 may facilitate osas#9 perception by interacting with another GPCR that directly binds to osas#9. Several studies have already demonstrated that GPCRs involved in ascaroside perception act as heterodimers[18], and it is possible that another receptor expressed in the ASH, ADL, and AWA neurons directly interacts with TYRA-2 and is responsible for binding osas#9. Furthermore, the neurotransmitter tyramine has been shown to activate the Gαi/o protein-coupled tyramine receptor, TYRA-2, in different sensory and interneurons to mediate different behaviors such as multisensory decision making and feeding suppression[57,58]. Tyramine released from the RIM interneurons activates the TYRA-2 receptor in ASH neurons to

mediate threat tolerance in a positive feedback loop[58]. Similarly, TYRA-2 functions in AIM interneurons to respond to tyramine release from the RIM interneurons in the mediation of feeding suppression[57]. Our results show that endogenous tyramine signaling is not directly involved in the response to osas#9, but instead other neurons and neuromodulatory signaling pathways participate in shaping the osas#9 response. Such modulation of the osas#9 response circuitry remains to be investigated.

Exactly how key innovations in metazoan signaling complexity evolved from pre-existing machineries remains to be elucidated[59]. Neurotransmitter signaling is typically inter-cellular, i.e., facilitating cell-to-cell communication, involves highly regulated biosynthesis of specific small molecules (e.g. biogenic amines), their translocation (either by way of diffusion or through active transport), and perception by dedicated receptors[60]. This mode of signaling is strikingly similar to pheromone-mediated communication systems, which rely on highly specific production and perception of small molecule ligands for inter-organismal signaling[61]. During evolution, it stands to reason that some machinery from inter-cellular signaling would also be utilized for inter-organismal signaling. Co-option of such signaling systems has been observed in both invertebrates (*C. elegans*), where a nicotinic acetylcholine receptor senses choline[62], and vertebrates (such as mice and zebrafish), where some metabotropic neurotransmitter receptors act as sensors to detect amino acids in the environment[63–65].

In summary, our findings demonstrate that the tyramine receptor, TYRA-2, functions in the chemosensation of osas#9, a neurotransmitter-derived inter-organismal signal. These results reveal the participation of both neurotransmitter biosynthesis and reception in inter-cellular as well as inter-organismal signaling. Hence, it appears that evolution of an inter-organismal communication pathway co-opted both a small molecule (octopamine), and a related receptor (TYRA-2) for mediating starvation-dependent dispersal in *C. elegans* (Fig. 7), suggesting that such co-option may represent one mechanism for the emergence of new inter-organismal communication pathways.

## Methods

**Avoidance drop test.** The tail end of a forward moving animal was subjected to a small drop (~5 nl) of solution, delivered through a hand-pulled 10 μl glass capillary tube. The solution, upon contact, was drawn up to the amphid sensory neurons via capillary action. In response, the animal either continued its forward motion (scored as "no avoidance response"), or displayed an avoidance response within 4 s[66]. The avoidance response was characterized a reversal if the behavior consisted of at least one half of a complete "head swing" followed by a change in direction of at least 90° from the original vector. For quantitative analysis, an avoidance response was scored as "1" and no response as "0". The avoidance index was calculated by dividing the number of avoidance responses by the total number of trials. Each trial was done concurrently with osas#9 and a solvent control.

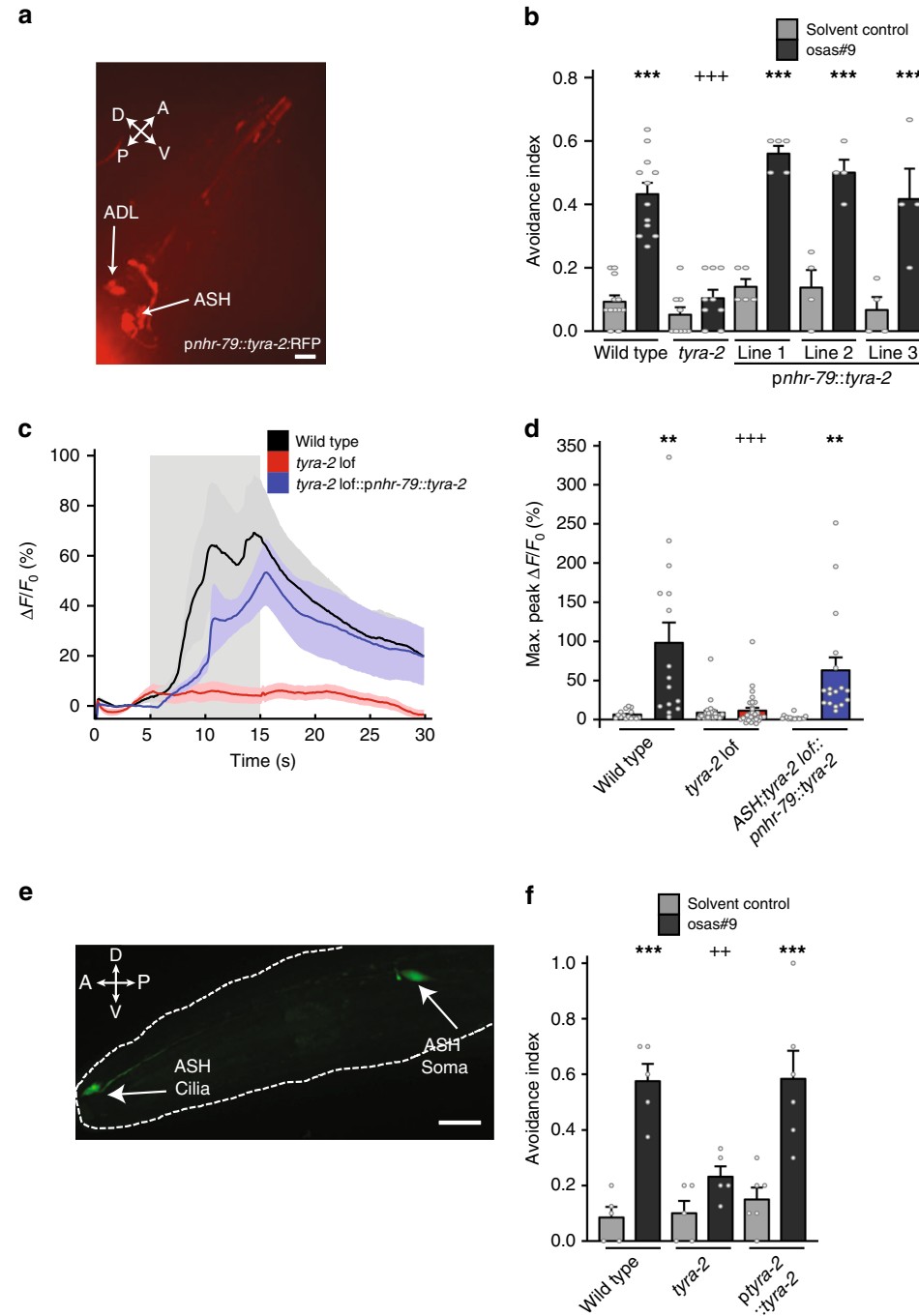

**Fig. 4** *tyra-2* expression in sensory neurons is required for sensitivity to osas#9. **a** A transcriptional rescue construct, p*nhr-79*::*tyra-2*::RFP, exhibits expression of *tyra-2* in both ASH and ADL neurons (×40 magnification, scale bar denotes 10 μm). **b** Rescue of *tyra-2* in ASH/ADL neurons fully reconstituted behavioral response to 1 μM osas#9. $n \geq 4$ trials. **c** Rescue of *tyra-2* in ASH neurons in a *tyra-2 lof* (blue) background show no difference compared to calcium transients of wild type animals expressing GCaMP3 in ASH neurons (black) when exposed to 1 μM osas#9 (gray-shaded region), while *tyra-2 lof* animals show a clear loss of calcium transcients in ASH (red), $n \geq 10$ animals. **d** Maximum fluorescence intensity in transgenic worms before (solvent control) and during exposure to 1 μM osas#9. **e** Sub-cellular localization of *tyra-2*. A translational reporter of the entire *tyra-2* genomic locus (p*tyra-2*::*tyra-2*::GFP) was injected into *tyra-2 lof* animals at 1 ng/μL, revealing expression of the receptor in both soma and sensory cilia. (60x magnification, scale bar denotes 20 μm). **f** Expression of the translational reporter restores wild type behavior in a *tyra-2 lof* background, $n \geq 5$ trials. Data presented as mean ± S.E.M; *$p < 0.05$, **$p < 0.01$, ***$p < 0.001$. One factor ANOVA with Sidak's multiple comparison posttest

Control animals and strains containing transgenes in various genetic backgrounds were prepared using common M9 buffer to wash and transfer a plate of animals to a microcentrifuge tube where the organisms are allowed to settle. The supernatant was removed and the animals were resuspended and allowed to settle again. The supernatant was again removed and the animals transferred to an unseeded plate. After 1 h, young adult animals were subjected to the solvent control

and chemical of interest at random, with no animal receiving more than one drop of the same solution. Refed animals were transferred to a seeded plate with M9 buffer, and after the allotted time, transferred to an unseeded plate and tested after 10 minutes.

Ablated and extrachromosomal transgenic animals and controls were gently passed onto an unseeded plate and allowed to crawl around. They were then again

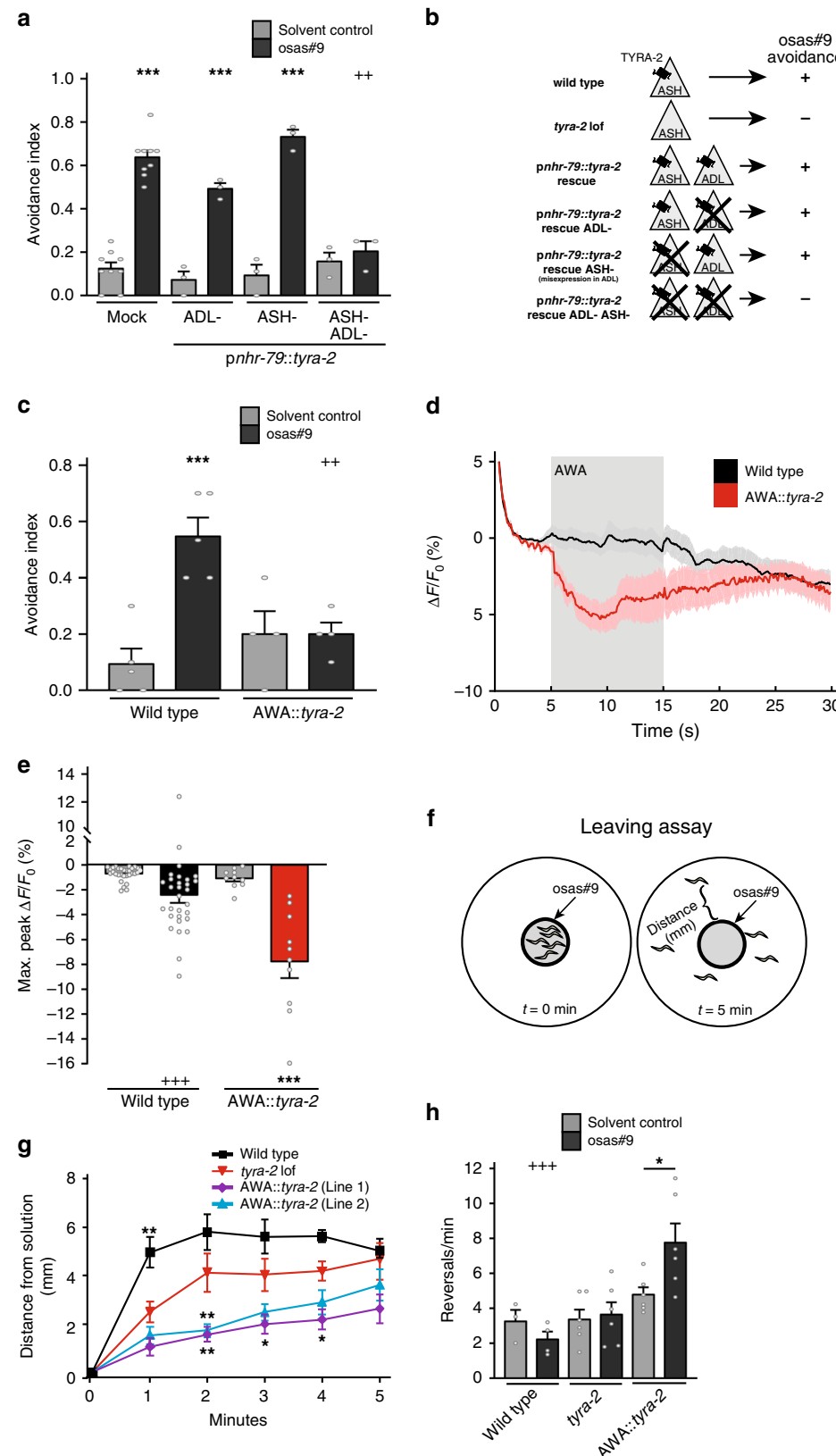

gently passed to another unseeded plate to minimalize bacterial transfer. Ablated animals were tested three times with the solvent control and solution of interest with 2 min intervals between drops[66].

**Strains and plasmids**. *tyra-2* rescue and misexpression plasmids were generated using MultiSite Gateway Pro Technology and injected into strain FX01846 *tyra-2*

(*tm1846*) with co-injection marker p*elt-2*;mCherry by Knudra Transgenics. The promoter attB inserts were generated using PCR and genomic DNA or a plasmid. The *tyra-2* insert was isolated from genomic DNA using attB5*ggcttatccgttgtggagaa* and attB2*ttggcccttccttttctctt*. PDONR221 p1-p5r and PDONR221 P5-P2 donor vectors were used with attB inserts. The resultant entry clones were used with the destination vector pLR305 and pLR306.

**Fig. 5** Ectopic expression of *tyra-2* confers the ability to respond to osas#9. **a** Misexpression of *tyra-2* in ADL neurons confers avoidance behavior in response to osas#9. *nhr-79* promoter driving *tyra-2* expression in ASH and ADL sensory neurons rescues osas#9 avoidance. Ablation of either ADL or ASH neurons does not affect osas#9 avoidance in the rescue lines, suggesting that misexpression of *tyra-2* in ADL neurons is sufficient for osas#9 response. Ablation of both ASH and ADL completely abolished avoidance. $n \geq 5$ trials. **b** Schematic illustration of cellular ablations in the transgenic rescue lines expressing *tyra-2* under the *nhr-79* promoter. **c** Animals with reprogrammed AWA sensory neurons in *tyra-2 lof* background do not avoid 1 μM osas#9. $n \geq 4$ trials. **d** AWA neurons (black) do not exhibit calcium transients in response to 1 μM osas#9, while reprogrammed AWA::*tyra-2* neurons (red) are hyperpolarized upon exposure (gray-shaded region), $n = 10$ animals. **e** Maximum fluorescence intensity in transgenic worms before (solvent control) and during exposure to 1 μM osas#9. **f** Schematic illustration of the leaving assay to measure osas#9 attraction. **g** Wild type (black), *tyra-2 lof* (red), and AWA::*tyra-2* lines (1 [magenta], 2 [cyan]) were subjected to 10 pM osas#9 in the leaving assay. Wild-type animals left the osas#9 solution spot quicker than the *tyra-2 lof* animals, whereas the misexpression lines remained closer to osas#9, $n \geq 3$ trials. **h** Reprogrammed AWA::*tyra-2* animals have an increased reversal rate in comparison to both wild type and *tyra-2 lof* animals in 10 pM osas#9, $n \geq 3$. Data presented as mean ± S.E.M; *$p < 0.05$, **$p < 0.01$, ***$p < 0.001$. One factor ANOVA with Sidak's multiple comparison posttest

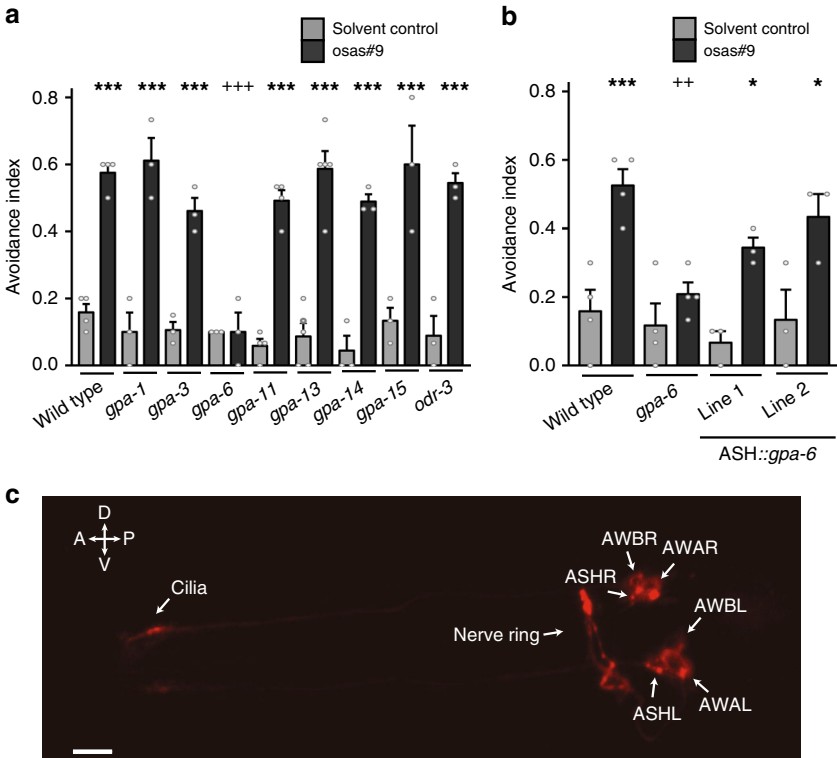

**Fig. 6** GPA-6 functions in ASH sensory neurons to mediate osas#9 response. **a** Screen of mutations in Gα subunits resulted in identification of the Gα subunit, *gpa-6*, which was defective in avoidance response to osas#9, $n \geq 3$ trials. **b** Expression of *gpa-6* in ASH neurons using *nhr-79* promoter reconstituted avoidance response similar to wild-type animals, $n \geq 3$ trials. **c** *gpa-6* localizes to the soma and cilia in ASH neurons. Translational fusion of the entire *gpa-6* genomic region displayed localization of the subunit to the soma of AWA, AWB, and ASH neurons. In addition, we also observed ciliary localization in ASH neurons (×40 magnification, scale bar denotes 10 μm). Data presented as mean ± S.E.M; *$p < 0.05$, **$p < 0.01$, ***$p < 0.001$. One factor ANOVA with Sidak's multiple comparison posttest

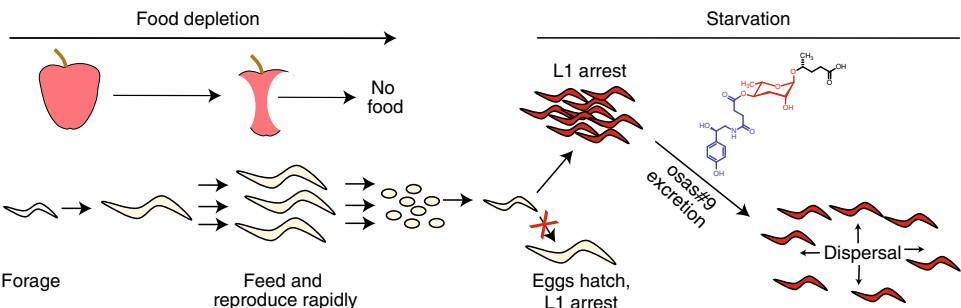

**Fig. 7** osas#9 serves as a dispersal cue for *C. elegans*. An animal navigating its environment encounters a food source, where offspring grow and reproduce rapidly, eventually depleting their food source. Eggs hatch on the depleted food patch and halt development as L1 arrest animals. L1 arrest animals secrete the aversive compound, osas#9, signaling for conspecifics to disperse away from the unfavorable condition

For AWA expression, a 1.2-kb *odr-10* promoter was isolated from genomic DNA using primers attB1*ctcgctaaccactcggtcat* and attB5r*gtcaactagggtaatccacaattc*. Entry clones were used with destination vector pLR305 resulting in p*odr-10::tyra-2::RFP* and co-injected with p*elt-2*::mCherry into FX01846.

For ASH expression, a 3-kb *nhr-79* promoter was isolated from genomic DNA using primers attB1*gtgcaatgcatggaaaattg* and attB5r*atacacttcccacgcaccat*. Entry clones were used with destination vector pLR306 resulting in p*nhr-79::tyra-2::RFP* and co-injected with p*elt-2*::mCherry into FX01846.

For ASH expression, a 3-kb *nhr-79* promoter was isolated from genomic DNA using primers attB1*gtgcaatgcatggaaaattg* and attB5r*atacacttcccacgcaccat*. *gpa-6* was isolated from genomic DNA using primers attB5 cgtctctttcgtttcaggtgtat and attB2 tattttcaaagcgaaacaaaaa. Entry clones were used with destination vector pLR304 resulting in p*nhr-79::gpa-6::RFP* and co-injected with p*unc-122*::RFP into NL1146.

*tyra-2*::GFP fusions were created by PCR fusion using the following primers to isolate 2 kb p*tyra-2* with its entire genomic locus from genomic DNA: A) atgtttttcacaagtttcaccaca, A nested) ttcacaagtttcaccacattacaa, and B with overhang) AGTCGACCTGCAGGCATGCAAGCT *gacacgagaagttgagctgggtttc*. GFP primers as described in WormBook[67]. The construct was then co-injected with p*elt-2*::mCherry into both N2 and FX01846.

*gpa-6*::RFP was generated by adding the restriction sites, AgeI and KpnI, to isolate 4 kb p*gpa-6* and the entire *gpa-6* locus from genomic DNA using primers: acatctggtacccctcaatttcccacgatct and acatctaccggtctcatgtaatccagcagacc. RFP::*unc-54*, ori, and AMPr was isolated from p*unc-122*::RFP plasmid by PCR addition of the restriction sites AgeI and KpnI with primers: acatctaccggt ATGGTGCGCT CCTCCAAG and ttaataggtacc*TGGTCATAGCTGTTTCCTGTG*. After digestion and ligation, the clone was injected into N2 with co-injection marker p*unc-122*::GFP.

(See Supplementary Tables 1–3 for details on strains, plasmids, and primers used in this study.)

**Chemical compounds**. ascr#3 and osas#9 were synthesized as previously described[10,13].

**RNA interference**. RNAi knockdown experiments were performed by following the RNAi feeding protocol found at Source Bioscience (https://www.sourcebioscience.com/products/life-sciences-research/clones/rnai-resources/c-elegans-rnai-collection-ahringer/). The RNAi clones (F01E11.5, F14D12.6, and empty pL4440 vector in HT115) originated from the Vidal Library[68], were generously provided by the Ambros Lab at UMASS Medical School. We observed that RNAi worked best when animals were cultured at 15 °C. We used VH624 (*nre-1* (hd20);*lin-15B*(hd126)) for the RNAi studies, as it has been previously shown to be sensitive to neuronal RNAi[28].

**Laser ablations**. Laser ablations were carried out using DIC optics and the MicroPoint laser system[69]. L1 worms were immobilized on 2% agarose on a glass slide using 1 mM sodium azide. The neurons of the L1 animals were identified and ablated at the nucleus by pulses of laser. Animals were removed from the slide and allowed to recover. Ablated animals were assayed 72 hours later, at the young adult stage. All ablated animals were tested in parallel with control animals that were treated similarly as ablated animals but were not exposed to the laser microbeam.

**Imaging**. Translational fusion animals were prepared for imaging by mounting them to a 4% agar pad with 10 mM levamisole on a microscope slide[70]. Animals were imaged using a Nikon Multispectral Multimode Spinning Disk Confocal Microscope, courtesy of Dr. Kwonmoo Lee at Worcester Polytechnic Institute or a Zeiss LSM700 Confocal Microscope, courtesy of the Department of Neurobiology at University of Massachusetts Medical School, Worcester.

Calcium imaging was perfomed by using a modified olfactory chip described in Reilly et al.[37]. For imaging, worms were treated in a similar way as the behavioral assays. Worms were starved for an hour when imaging with osas#9. A young adult animal was immobilized in a PDMS olfactory chip with its nose subject to a flowing solution. Animals were imaged at ×40 magnification for 30 s, and experienced a 10 s pulse of osas#9, tyramine, octopamine, or ascr#3 in between the solvent control. Each animal was exposed to the stimulus up to three times; multiple exposures to the chemical did not show a significant difference in response between exposures. Soma fluorescence from GCaMP3 was measured using ImageJ. Background subtraction was performed for each frame to obtain the value $\Delta F$. Change in fluorescence ($\Delta F/F_0$) was calculated by dividing the $\Delta F$ value of each frame by $F_0$. $F_0$ was calculated as the average $\Delta F$ of 10 frames prior to stimulus exposure[37]. $\Delta F/F_0$ (%) was calculated by subtracting 1 from $\Delta F/F_0$ and multiplying 100%; these calculations were then plotted over the duration of the experiment.

**Quantitative RT-qPCR**. RNA was isolated from individual animals, either freshly removed from food or after four hours of starvation using Proteinase K buffer[71]. cDNA was subsequently synthesized using the Maxima H Minus First Strand cDNA Synthesis Kit. iTaq Universal SYBR Green Supermix was used for amplification with the Applied Biosystem 7500 Real Time system. Primer efficiency was determined to be 97.4% for *tyra-2* primers (GAGGAGGAAGAAGATAGC GAAAG, TGTGATCATCTCGCTTTTCA) and 101.8% for the reference gene

*ama-1* (GGAGATTAAACGCATGTCAGTG, ATGTCATGCATCTTCCACGA) using the equation 10^(-1/slope)-1. Technical replicates with large standard deviations and trials with a Ct within 5 cycles of the negative control (no reverse transcriptase used in prep) were removed from analyses.

**Locomotion**. Speed*:* Five animals were gently transferred to a 35-mm plate and filmed for 20 minutes. Movies were generated using the Wormtracker system by MBF Bioscience. Movies were then analyzed and average speed was computed using software WormLab4.1 (MBF Bioscience, Williston, VT).

Reversals: reversals were analyzed and measured using Wormlab (MBF Bioscience) from movies recorded for the holding assay between minute one and two as it was when the divergence was first seen in distance between strains in the holding assay.

**Chemoattraction**. Diacetyl chemotaxis assays were carried out with slight modifications[72]. 10 animals were placed in the center of a 35-mm plate, equidistant from two spots, one containing 1 µl of solvent control and the other 1 µl of $10^{-2}$ diacetyl. Both spots contained sodium azide for anesthetizing animals that entered the region. After 45 minutes, the chemotaxis index was calculated by subtracting the number of animals in the solvent control from the number of animals in the solution of interest and divided by the total number of animals.

**Leaving assay**. The leaving assay consisted of the use of 60 mm culture plates containing standard NGM agar. A transparency template that included a 6-mm diameter circle in the center was attached to the underside of the NGM plate. One hour before running the assay, young adult animals were passed on to an unseeded plate and allowed to starve for one hour. 100 µl of *E. coli* OP50 liquid culture was spread onto a separate NGM assay plates. These plates were allowed to dry at 25 °C without a lid for one hour. After an hour of incubation, 4 µl of either solvent control or 10 pM osas#9 was pipetted onto the agar within the center circle outlined on the template. Ten animals were gently passed into the center circle and their movement was recorded. At 1-minute intervals, the distance the animals traveled from the origin was measured using ImageJ.

**Statistical analyses**. Statistical tests were run using Graphpad Prism. For all figures, when comparing multiple groups, one factor ANOVAs were performed, followed by Sidak's multiple comparison test. When only two groups were compared, a Student's *t*-test was used. All tests were two-tailed. When comparing different strains/conditions, normalized values of osas#9 avoidance index response relative to the respective solvent control were used. This was done to account for any differences in baseline response to solvent control for the respective genotypes, laser ablations, or physiological conditions. When normalizing fold change of osas#9 response to solvent control response for the avoidance assay within a strain/condition, data was first log transformed so a fold change could still be calculated for control plates having a "0" value. For avoidance assays, statistical groups were based on the number of plates assayed, not the number of drops/animals. For calcium imaging, averages were calculated by obtaining the max peak value before and during exposure to the chemical of interest for each trial.

For all figures, asterisks depict compared osas#9 avoidance to respective solvent control within groups. ' + ' signs represent same *p*-value as asterisks, but represent the difference between osas#9 avoidance of a strain/conditions in comparison to wild type, with the exception of panel 5H, which shows the difference between response of all strains/conditions and the reprogrammed AWA::*tyra-2*

**Reporting summary**. Further information on research design is available in the Nature Research Reporting Summary linked to this article.

## Data availability
The authors declare that all data supporting the findings of this study are available within the paper and its supplementary information files. The source data underlying all figures (Fig. 1–6, Supplementary Fig. 1–6) are provided as a Source Data file.

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

## Acknowledgements

We thank the Caenorhabditis Genetics Center (CGC), which is funded by the NIH Office of Research Infrastructure Programs (P40 OD010440), R. Komuniecki, S. Suo, D. Chase, V. Ambros, C. Bargmann, E.M. Schwarz, and P. Sternberg for strains; R. Garcia, D. Albrecht, and S. Chalasani for plasmids; Knudra transgenics and W. Joyce for injections; K. Lee for the use of the spinning-disk confocal microscope; UMMS Neurobiology department and M. Gorczyca for assistance and use of confocal microscope; V. Ambros, Dana-Farber Cancer Institute, and BioScience Life Sciences for Vidal library RNAi clones; A. Maurya and Piali Sengupta for technical suggestions; D. Vargas Blanco for RT-qPCR guidance; the Srinivasan lab, Rick Komuniecki, Michael Nitabach and Nitabach lab and S. Chalasani for critical comments on the manuscript; A. Warty for contribution to glycerol assays. This work was supported in by grants from the NIH (R01DC016058 to J.S. and GM113692 and GM088290 to F.C.S. and GM084491 to M.J.A.).

## Author contributions

C.D.C. performed the molecular biology, ablations, and behavioral assays. C.D.C. and E.M.D. performed calcium imaging. D.K.R. performed the competition experiments, C.D.C. and V.L.C. performed the RNAi behavioral assays. Y.K.Z. synthesized osas#9. H.J.C. helped in confocal microscopy of transgenic strains. D.R. generated strains from MJA lab. C.D.C., E.M.D., D.K.R., and J.S. wrote the manuscript, with input from F.C.S. and M.J.A.

## Additional information

**Competing interests:** The authors declare no competing interests.

**Peer Review Information:** *Nature Communications* thanks the anonymous reviewer(s) for their contribution to the peer review of this work. Peer reviewer reports are available.

