## [Peer Review File · Nature Communications]

Reviewers' comments:

Reviewer #1 (Remarks to the Author):

Chute and colleagues present evidence that the tyramine receptor TYRA-2 functions as a sensory receptor for the octopamine-derived *osas#9* pheromone in *C. elegans*. This is a very intriguing discovery that illustrates the co-option of a ligand and receptor that normally function in inter-neuronal communication as a signaling mechanism between individual animals. The evidence for these claims are largely very convincing and the figures and text clear. I have only a few comments:

- there are no figure legends in the submitted file. Fortunately, the figures are quite intuitive, although I was unclear about meaning of the asterisks/plus symbol indicating statistical differences. (Hence it is difficult to provide a comment about statistical analyses).

- can the authors rescue the *osas#9*-evoked calcium responses in the mutant background with a wild-type transgene (as they do for the behavioral experiments)?

- the authors show that mis-expression of TYRA-2 in AWA neurons changes the behavioral response to *osas#9*, from which they conclude that TYRA-2 itself is involved in pheromone sensing. This proposition would be greatly strengthened if they perform calcium imaging in AWA to determine whether the neurons now display *osas#9*-evoked calcium responses (I wouldn't consider this an essential experiment, but it seems like they have all the tools available). If this didn't work in AWA – and I appreciate signal transduction pathway are different in distinct neuron in *C. elegans* – other heterologous neuron types could be tested, or a cell culture system.

- can the authors clarify why expression of TYRA-2 in AWA neurons does not lead to attraction per se (rather than a reduced tendency to exit the zone around a source of ligand in the “leaving assay”).

- do *gpa-6* mutants display a similar neuro-physiological defect in ASI in response to *osas#9* as *tyra-2* mutants? (This is not an essential experiment).

- page 3 – the authors describe chemical communication as “less apparent”, which I assume means “less apparent to humans”. I don't think this contrast is really needed.

- page 4 – the authors indicate co-option of intracellular signaling has been “hinted at” before. This is slightly unfair to the mammalian examples given, and also overlooks other cases when intra-organismal signaling receptors function as sensory receptors, albeit not necessarily in pheromone sensing (e.g., the invertebrate Ionotropic Receptors, which have derived from iGluRs or environmental choline sensing in *C. elegans* by acetylcholine receptors; PMID 20029439 provides a nice review of this topic.)

- Figure 4D – the authors write “Additionally, injection of the *tyra-2* translational reporter into *tyra-2* *lof* animals displayed sub-cellular localization in the ASH sensory cilia”, which isn't scientifically/grammatically correct. I also wondered whether the reporter localized to cilia in wt animals too (it is not necessary to do this experiment; I realize the interest of doing it in the mutant background was to shown rescue at the same time as looking at protein localization).

- Discussion – this could be shortened a bit (it's currently about half the length of the Results). Greater concision would help the main messages stand out by avoiding repetition.

Reviewer #2 (Remarks to the Author):

Chute et al. report the discovery of a novel function for a *C. elegans* G protein-coupled receptor that was previously thought to function exclusively as a receptor for monoamine neurotransmitters. TYRA-2 was so named because it can be activated in vitro by tyramine and octopamine, which are invertebrate norepinephrine analogs. Chute and colleagues report that TYRA-2 is expressed in sensory neurons and mediates their activation by a pheromone that consists of a glycolipid derivatized with an octopamine moiety, *osas#9*. Nociceptor neurons that express TYRA-2 are activated by this pheromone at micromolar concentrations, and forced expression of TYRA-2 in other sensory neurons can confer pheromone-response behaviors. The authors also identify a specific G protein that is required for pheromone-sensing by ASH.

The study is logical and thorough with respect to establishing neural and molecular mechanisms required to sense the *osas#9* pheromone. The pheromone itself is an interesting molecule because of its likely ethological importance; *osas#9* is released by starved juveniles and triggers their dispersal. The authors' main conclusions - that TYRA-2 is a receptor for *osas#9* and that this might be an example of a neurotransmitter receptor co-opted for pheromone sensing - are supported by the data and will be of interest to many. The manuscript should be revised to address some of the following issues:

1. The authors rigorously demonstrate that *tyra-2* mutation affects *osas#9*-sensing using multiple alleles, transgene rescue, and RNAi. However, *osas#9* responses are only tested at one concentration (1 micromolar). This was done for good reason; one micromolar is the EC50 for eliciting avoidance. The authors should test higher concentrations of *osas#9* to determine whether there exist other low-affinity *osas#9* receptors, something that is suggested by their observation that *osas#9* avoidance is not completely eliminated by *tyra-2* mutation.
2. I would encourage the authors to test whether ASH neurons respond to other ascaroside pheromones in a TYRA-2-dependent manner. This is not an essential point, but the data in this manuscript suggest a high-affinity *osas#9*-sensing mechanism resident to ASH and I am curious to know whether ASHs sense other pheromones.
3. The authors nicely demonstrate that *osas#9* activates ASH neurons in a TYRA-2-dependent manner, and they further show that tyramine and octopamine are poor agonists of ASH neurons. I encourage the authors to include their studies of behavioral and cell-physiological responses to octopamine and tyramine into main figures. These data argue against a model in which TYRA-2 acts as a neurotransmitter receptor to modulate sensory neuron function and strengthen the authors' model, in which TYRA-2 is a high-affinity chemoreceptor.
4. The authors perform two experiments to test whether expression of TYRA-2 is sufficient for *osas#9* sensing. In one experiment TYRA-2 is expressed in AWAs, which mediate attraction. Here, the authors show that forced expression of TYRA-2 in AWAs causes attraction to *osas#9*. This is clearly explained. In another experiment, TYRA-2 is expressed in ASH neurons, which normally mediate *osas#9* avoidance and are necessary for avoidance, and ADL neurons, which mediate avoidance but do not normally express TYRA-2. In these transgenics, *osas#9* avoidance is no longer ASH-dependent, indicating that TYRA-2 expression in ADL can complement loss of the normal *osas#9*-sensing neurons. This is a good experiment and a nice result. I encourage the authors to illustrate the logic, perhaps with a schematic, to ensure that readers get the point.
5. Because TYRA-2 expression in ADLs and AWAs supports behavioral responses to *osas#9*, I expect

that expression of TYRA-2 also confers upon these cells the ability to display a physiological response to osas#9. Have the authors tested this? To be clear - I do not believe that the authors' model demands that they observe specific cellular responses in transgenic AWAs and ADLs. A naive expectation is that transgenic neurons show calcium responses to pheromone, but it is possible that cellular responses are not readily measured by calcium imaging. However, if the authors have attempted such experiments, I encourage them to describe them. We are familiar with chemoreceptors that are sufficient to confer cellular responses to sensory neurons that are readily observed with calcium indicators. The community would benefit from knowing if TYRA-2 acts in a more complicated manner, and for this reason even negative results would have value.

6. The summary figure is a bit of a mess and should be revised. First, there is a factual error. TDC-1 does not function as indicated in panel B. It is the decarboxylase that makes tyramine from tyrosine. TBH-1 is the hydroxylase. Second, I am unsure what point the authors want to emphasize. There is the ethological relevance of osas#9, which is illustrated in panel A, and the biosynthesis of osas#9 in panel B. But somehow the figure fails to emphasize the point that the authors repeatedly make in the manuscript: TYRA-2 is a monoamine receptor that also functions as a chemoreceptor. Please improve this figure or remove it.

Minor issues

7. The authors want to create a distinction between inter-cellular communication and inter-organism communication. They sometime refer to intra-cellular communication, which is confusing - this is probably a typo.

8. Throughout the manuscript and in many figures the authors use 'wildtype' when they should refer to the wild type or use 'wild-type' as a compound adjective.

Reviewer #3 (Remarks to the Author):

The manuscript entitled "Co-option of neurotransmitter signaling for inter-organismal communication in *C. elegans*" by Christopher Chute and coworkers reports a very interesting phenomenon in nature, where a neurotransmitter derivative acts as an inter-organismal signaling molecule recognized by a tyramine receptor TYRA-2. The authors established an avoidance behavior assay for testing biological activity for ascaroside osas#9 which incorporates the neurotransmitter octopamine. The authors show that ASH sensory neurons and TYRA-2 expression in ASH neurons are required for this avoidance behavior using neuronal ablation, cell-specific genetic rescue, and calcium imaging. The authors also show that TYRA-2 ectopically expressed in AWA neurons is sufficient to turn these osas#9 insensitive neurons to osas#9 sensitive neurons. The experiment data here support their conclusion that TYRA-2 is required for sensing osas#9, and is probably a receptor for osas#9. Overall, this is an interesting study. I have some comments which I hope the authors will be able to address:

1) TYRA-2 has been shown to have high affinity ($K_d=20$ nM) toward tyramine in vitro. However, only up to 1 mM tyramine can induce aversive behavior and ASH neuronal calcium response, whereas μ M range of osas#9 is sufficient. There is a discrepancy here. Do tyramine and osas#9 bind to the same receptor? First, the authors need to check if tyra-2 lof mutation also abolishes calcium and behavior response for tyramine. Second, the authors need to do a competitive experiment between osas#9 and tyramine in both behavior and calcium imaging assays. The authors can mix varying amounts of osas#9 and tyramine to see whether and how such mixing changes biological potency. Lastly, the authors may also want to let the worm adapted to one chemical and then check worms' response to the other. These experiments should provide some clues.

- 2) Along the same line, does *osas#9* directly bind and activate TYRA-2? This can be tested in cultured cells in vitro.
- 3) Does *tyra-2* mRNA level increase under starved condition underlie *osas#9* sensitivity? Does the mRNA level of *tyra-2* correspond to worms' sensitivity to *osas#9*?
- 4) Did the authors also let the worm starved for 1h before doing calcium imaging for *osas#9*, or instead the authors used well-fed worms? The authors need to clarify their condition, and how this imaging data corresponds to their behavior data.
- 5) Do calcium signals arise from ASH or from other neurons? The authors need to perform calcium imaging in *unc-13* and *unc-31* mutant background.
- 6) Page 6, graph 3, first sentence, "*osas-9*" should be "*osas#9*".

Response to reviewers

Reviewer #1:

Reviewer Comment-1: Chute and colleagues present evidence that the tyramine receptor TYRA-2 functions as a sensory receptor for the octopamine-derived osas#9 pheromone in C. elegans. This is a very intriguing discovery that illustrates the co-option of a ligand and receptor that normally function in inter-neuronal communication as a signaling mechanism between individual animals. The evidence for these claims are largely very convincing and the figures and text clear. I have only a few comments:

- there are no figure legends in the submitted file. Fortunately, the figures are quite intuitive, although I was unclear about meaning of the asterisks/plus symbol indicating statistical differences. (Hence it is difficult to provide a comment about statistical analyses).

Author response: We apologize for this mistake as the main manuscript file did not integrate the figure captions. The revised version of the manuscript now incorporates the figure legends for all the figures within the manuscript.

Reviewer Comment-2 - can the authors rescue the osas#9-evoked calcium responses in the mutant background with a wild-type transgene (as they do for the behavioral experiments)?

Author response: We generated a new transgenic line (JSR104), wherein we rescue *tyra-2* in the ASH neuron. Imaging this line (Fig. 4C,D) shows rescue of the calcium transients upon osas#9 stimulation in the ASH neuron.

Reviewer Comment-3 - the authors show that mis-expression of TYRA-2 in AWA neurons changes the behavioral response to osas#9, from which they conclude that TYRA-2 itself is involved in pheromone sensing. This proposition would be greatly strengthened if they perform calcium imaging in AWA to determine whether the neurons now display osas#9-evoked calcium responses (I wouldn't consider this an essential experiment, but it seems like they have all the tools available). If this didn't work in AWA – and I appreciate signal transduction pathway are different in distinct neuron in C. elegans – other heterologous neuron types could be tested, or a cell culture system.

Author response: We generated a transgenic line (JSR45) in which we express *tyra-2* under an AWA-specific promoter (*odr-10*). Upon stimulating this line with osas#9, we observe hyperpolarizing calcium transients in the AWA neuron (Fig. 5D,E), further supporting involvement of TYRA-2 itself in pheromone sensing.

Reviewer Comment-4 - can the authors clarify why expression of TYRA-2 in AWA neurons does not lead to attraction per se (rather than a reduced tendency to exit the zone around a source of ligand in the “leaving assay”).

Author response: Olfactory sensory neurons can be reprogrammed by expressing specific receptors to change the valence of an odorant chemical (Troemel et al., Cell, 1997). Our studies suggest that mis-expression of TYRA-2 in the AWA neurons, which are typically known to mediate attractive behaviors, results in slower dispersal compared to wild type animals. We believe that expressing just the TYRA-2 receptor in AWA neurons is insufficient to mediate complete attraction, as it may require additional signaling components within these AWA neurons. However, the results from the leaving assay clearly suggest that TYRA-2 confers the ability to sense osas#9,

and this results in attraction-like responses. Alternatively, TYRA-2 receptor may activate different signaling pathways compared to traditional chemosensory signaling pathways within neurons.

Reviewer Comment-5 - do gpa-6 mutants display a similar neuro-physiological defect in ASI in response to osas#9 as tyra-2 mutants? (This is not an essential experiment).

Author response: We thank the reviewer for their valuable suggestion, as this would be an interesting avenue of investigation. We are planning to study the role of *gpa-6* and additional signaling components for their role in tyramine-mediated avoidance behavior. However, we believe that the physiological response of ASI to *osas#9* is beyond the scope of the present manuscript.

Reviewer Comment-6: - page 3 – the authors describe chemical communication as “less apparent”, which I assume means “less apparent to humans”. I don’t think this contrast is really needed.

Author response: We have edited the text to allow for better clarity.

Reviewer Comment-7: - page 4 – the authors indicate co-option of intracellular signaling has been “hinted at” before. This is slightly unfair to the mammalian examples given, and also overlooks other cases when intra-organismal signaling receptors function as sensory receptors, albeit not necessarily in pheromone sensing (e.g., the invertebrate Ionotropic Receptors, which have derived from iGluRs or environmental choline sensing in C. elegans by acetylcholine receptors; PMID 20029439 provides a nice review of this topic.)

Author response: We are grateful to the reviewer for this excellent suggestion and apologize for the oversight in citing relevant work. The revised version of our manuscript now discusses and cites relevant research showing intra-organismal signaling receptors functioning as sensory receptors in different mammalian systems.

Reviewer Comment-8: - Figure 4D – the authors write “Additionally, injection of the tyra-2 translational reporter into tyra-2 lof animals displayed sub-cellular localization in the ASH sensory cilia ...”, which isn’t scientifically/grammatically correct. I also wondered whether the reporter localized to cilia in wt animals too (it is not necessary to do this experiment; I realize the interest of doing it in the mutant background was to shown rescue at the same time as looking at protein localization).

Author response: We corrected this sentence. We had originally injected the translational reporter in wild type worms at 30 ng/μl to check for cellular localization of the protein. This concentration of the construct resulted in saturation of the protein in the soma and dendrites, and we were not able to observe any cilia expression (Fig. 3A). To characterize sub-cellular localization of the protein, we used a lower concentration of the construct (1 ng/μl) in the *tyra-2* lof background (Fig. 4G). Since the only difference between the two strains is a deletion of the receptor being rescued, we do not expect there to be any difference in the localization of TYRA-2 protein in wild type animals compared to the transgenic worms expressing the translational reporter. We also performed behavioral analyses in the rescued lines and demonstrate that we are able to rescue the avoidance behavior in the transgenic line (Fig. 4F).

Reviewer Comment-9: - Discussion – this could be shortened a bit (it's currently about half the length of the Results). Greater concision would help the main messages stand out by avoiding repetition.

Author response: Upon reviewing our original manuscript, we concur with the reviewer that the discussion can be shortened. The revised version of the manuscript aims to focus on the main implications of our results.

Reviewer #2 (Remarks to the Author):

Chute et al. report the discovery of a novel function for a C. elegans G protein-coupled receptor that was previously thought to function exclusively as a receptor for monoamine neurotransmitters. TYRA-2 was so named because it can be activated in vitro by tyramine and octopamine, which are invertebrate norepinephrine analogs. Chute and colleagues report that TYRA-2 is expressed in sensory neurons and mediates their activation by a pheromone that consists of a glycolipid derivatized with an octopamine moiety, osas#9. Nociceptor neurons that express TYRA-2 are activated by this pheromone at micromolar concentrations, and forced expression of TYRA-2 in other sensory neurons can confer pheromone-response behaviors. The authors also identify a specific G protein that is required for pheromone -sensing by ASH.

The study is logical and thorough with respect to establishing neural and molecular mechanisms required to sense the osas#9 pheromone. The pheromone itself is an interesting molecule because of its likely ethological importance; osas#9 is released by starved juveniles and triggers their dispersal. The authors' main conclusions - that TYRA-2 is a receptor for osas#9 and that this might be an example of a neurotransmitter receptor co-opted for pheromone sensing - are supported by the data and will be of interest to many. The manuscript should be revised to address some of the following issues:

Reviewer Comment-1: The authors rigorously demonstrate that tyra-2 mutation affects osas#9-sensing using multiple alleles, transgene rescue, and RNAi. However, osas#9 responses are only tested at one concentration (1 micromolar). This was done for good reason; one micromolar is the EC50 for eliciting avoidance. The authors should test higher concentrations of osas#9 to determine whether there exist other low-affinity osas#9 receptors, something that is suggested by their observation that osas#9 avoidance is not completely eliminated by tyra-2 mutation.

Author response: We thank the reviewer for their positive comments on our manuscript. In our revised version, we have tested several higher concentrations of osas#9 (1 μ M, 10 μ M and 100 μ M) on wild type and two different tyra-2 loss of function strains (Fig. 2D). We observe that lack of the TYRA-2 receptor results in loss of avoidance at both 1 μ M and 10 μ M. However, at 100 μ M, we still observe strong avoidance (Fig. 2D). Avoidance behavior observed at this non-physiologically relevant concentration suggests that other low-affinity (possibly less or non specific) receptors are involved.

Reviewer Comment- 2: I would encourage the authors to test whether ASH neurons respond to other ascaroside pheromones in a TYRA-2-dependent manner. This is not an essential point, but the data in this manuscript suggest a high-affinity osas#9-sensing mechanism resident to ASH and I am curious to know whether ASHs sense other pheromones.

Author response: We agree that this is an interesting question. In the revised version of our manuscript, we have now tested an unrelated ascaroside, ascr#3. This small-molecule has been previously shown to be necessary for mediating male attraction. Exposure of worms to this chemical did not elicit any calcium transients, suggesting that ASH neurons respond specifically to osas#9 stimulation (Supplementary Fig. 3 E,F).

Reviewer Comment-3: The authors nicely demonstrate that osas#9 activates ASH neurons in a TYRA-2-dependent manner, and they further show that tyramine and octopamine are poor

agonists of ASH neurons. I encourage the authors to include their studies of behavioral and cell-physiological responses to octopamine and tyramine into main figures. These data argue against a model in which TYRA-2 acts as a neurotransmitter receptor to modulate sensory neuron function and strengthen the authors' model, in which TYRA-2 is a high-affinity chemoreceptor.

Author response: We now include the behavioral and physiological data for tyramine in the main figures (Fig. 3 G-I). The octopamine results have been kept in the supplement file (Supplementary Fig. 4 A-C) for space reasons.

Reviewer Comment-4: The authors perform two experiments to test whether expression of TYRA-2 is sufficient for osas#9 sensing. In one experiment TYRA-2 is expressed in AWAs, which mediate attraction. Here, the authors show that forced expression of TYRA-2 in AWAs causes attraction to osas#9. This is clearly explained. In another experiment, TYRA-2 is expressed in ASH neurons, which normally mediate osas#9 avoidance and are necessary for avoidance, and ADL neurons, which mediate avoidance but do not normally express TYRA-2. In these transgenics, osas#9 avoidance is no longer ASH-dependent, indicating that TYRA-2 expression in ADL can complement loss of the normal osas#9-sensing neurons. This is a good experiment and a nice result. I encourage the authors to illustrate the logic, perhaps with a schematic, to ensure that readers get the point.

Author response: We thank the reviewer for their suggestion. The revised Fig. 4 now has a schematic describing the logic of the ADL mis-expression experiment (Fig. 5B).

Reviewer Comment-5: Because TYRA-2 expression in ADLs and AWAs supports behavioral responses to osas#9, I expect that expression of TYRA-2 also confers upon these cells the ability to display a physiological response to osas#9. Have the authors tested this? To be clear - I do not believe that the authors' model demands that they observe specific cellular responses in transgenic AWAs and ADLs. A naïve expectation is that transgenic neurons show calcium responses to pheromone, but it is possible that cellular responses are not readily measured by calcium imaging. However, if the authors have attempted such experiments, I encourage them to describe them. We are familiar with chemoreceptors that are sufficient to confer cellular responses to sensory neurons that are readily observed with calcium indicators. The community would benefit from knowing if TYRA-2 acts in a more complicated manner, and for this reason even negative results would have value.

Author response: We have conducted experiments that attempt to measure calcium transients in neurons mis-expressing *tyra-2* in the AWA neurons. As described in our original submission, we generated a transgenic line (JSR45) that expresses *tyra-2* under an AWA-specific promoter (*odr-10*), and this mis-expression switched the valence of the response, from repulsion to attraction (Fig. 5 C). In the revised version of the manuscript, we have added results from *in vivo* imaging analysis of AWA neurons, showing distinct hyperpolarization in AWA::*tyra-2* animals upon osas#9 stimulation, whereas no change in fluorescence was observed in wild type animals lacking *tyra-2* in the AWA neurons (Fig. 5 D,E). Both the behavioral experiments and the neurophysiological responses in AWA neurons thus indicate that expression of TYRA-2 confers onto the neuron the ability to detect osas#9.

Reviewer Comment-6: The summary figure is a bit of a mess and should be revised. First, there is a factual error. TDC-1 does not function as indicated in panel B. It is the decarboxylase that makes tyramine from tyrosine. TBH-1 is the hydroxylase. Second, I am unsure what point the authors want to emphasize. There is the ethological relevance of osas#9, which is illustrated in panel A, and the biosynthesis of osas#9 in panel B. But somehow the figure fails to emphasize

the point that the authors repeatedly make in the manuscript: TYRA-2 is a monoamine receptor that also functions as a chemoreceptor. Please improve this figure or remove it.

Author response: We apologize for the factual error in our summary figure and agree with the reviewer's point about the potential of Panel B. In our revised version, we have removed the panel and only emphasize the ethological relevance of the chemical.

Minor issues

Reviewer Comment-7: The authors want to create a distinction between inter-cellular communication and inter-organism communication. They sometime refer to intra-cellular communication, which is confusing - this is probably a typo.

Author response: We have corrected the typo in the revised version.

Reviewer Comment-8: Throughout the manuscript and in many figures the authors use 'wildtype' when they should refer to the wild type or use 'wild-type' as a compound adjective.

Author response: We have changed the usage of the word "wild type" to be consistent throughout the manuscript.

Reviewer #3 (Remarks to the Author):

The manuscript entitled “Co-option of neurotransmitter signaling for inter-organismal communication in C. elegans” by Christopher Chute and coworkers reports a very interesting phenomenon in nature, where a neurotransmitter derivative acts as an inter-organismal signaling molecule recognized by a tyramine receptor TYRA-2. The authors established an avoidance behavior assay for testing biological activity for ascaroside osas#9 which incorporates the neurotransmitter octopamine. The authors show that ASH sensory neurons and TYRA-2 expression in ASH neurons are required for this avoidance behavior using neuronal ablation, cell-specific genetic rescue, and calcium imaging. The authors also show that TYRA-2 ectopically expressed in AWA neurons is sufficient to turn these osas#9 insensitive neurons to osas#9 sensitive neurons. The experiment data here support their conclusion that TYRA-2 is required for sensing osas#9, and is probably a receptor for osas#9. Overall, this is an interesting study. I have some comments which I hope the authors will be able to address:

*Reviewer Comment-1: TYRA-2 has been shown to have high affinity ($K_d=20$ nM) toward tyramine in vitro. However, only up to 1 mM tyramine can induce aversive behavior and ASH neuronal calcium response, whereas μ M range of osas#9 is sufficient. There is a discrepancy here. Do tyramine and osas#9 bind to the same receptor? First, the authors need to check if *tyra-2* lof mutation also abolishes calcium and behavior response for tyramine. Second, the authors need to do a competitive experiment between osas#9 and tyramine in both behavior and calcium imaging assays. The authors can mix varying amounts of osas#9 and tyramine to see whether and how such mixing changes biological potency. Lastly, the authors may also want to let the worm adapted to one chemical and then check worms' response to the other. These experiments should provide some clues.*

Author response: To address whether loss of *tyra-2* abolishes calcium responses for tyramine, we generated a *tyra-2* lof transgenic line expressing the calcium sensor GCaMP in the ASH sensory neuron (JSR100). We observed that *tyra-2* lof mutation abolishes calcium responses in ASH neuron in response to 1 mM tyramine (Fig. 3H-I). We have also performed competition assays between tyramine and osas#9 in both wild type and *tyra-2* lof animals. In these experiments, we varied the concentration of tyramine relative to a constant concentration of osas#9 in the mixture. The additional presence of tyramine did not affect the response to osas#9 at equimolar concentrations, however we observed avoidance at non-physiological concentrations of tyramine in both wild type and *tyra-2* lof animals, suggesting that these responses are TYRA-2 independent (Supplementary Fig. 4D). Tyramine has been shown to bind to different receptors including *tyra-2*, *tyra-3*, *ser-2*, and *ser-3*, which could be involved. Moreover, millimolar concentrations of tyramine may simply elicit non-specific avoidance responses to basic amines.

Reviewer Comment-2: Along the same line, does osas#9 directly bind and activate TYRA-2? This can be tested in cultured cells in vitro.

Author response: We believe that our genetic, neurophysiological and mis-expression analyses suggests that the *tyra-2* receptor is sufficient to confer sensation of the cue and elicit a behavioral response. However, we cannot exclude that osas#9 perception involved interaction of TYRA-2 with another receptor, and previous work on ascaroside receptors demonstrates that in some heterodimers of GPCRs are involved in perception some ascarosides (Park et al, PNAS, 2012). We are currently collaborating with the Forrester Lab at the University of Ontario, Canada, to test

the binding of osas#9 and TYRA-2 in *Xenopus* oocytes. If we observe a lack of direct binding, this could be attributed to the need of heterodimeric receptors needed to sense osas#9.

Reviewer Comment-3: Does tyra-2 mRNA level increase under starved condition underlie osas#9 sensitivity? Does the mRNA level of tyra-2 correspond to worms' sensitivity to osas#9?

Author response: We thank this reviewer for this suggestion. RT-qPCR analyses (Fig. 2E) revealed a nearly two-fold increase in *tyra-2* expression in starved worms compared to well-fed animals, consistent with the idea that *tyra-2* mRNA levels in part regulate response to osas#9.

Reviewer Comment-4: Did the authors also let the worm starved for 1h before doing calcium imaging for osas#9, or instead the authors used well-fed worms? The authors need to clarify their condition, and how this imaging data corresponds to their behavior data.

Author response: For our imaging experiments, the worms were treated in a similar manner as in our behavioral experiments, including starvation for 1 hour prior to exposure to osas#9 exposure. We have added this information to the Methods section.

Reviewer Comment-5: Do calcium signals arise from ASH or from other neurons? The authors need to perform calcium imaging in unc-13 and unc-31 mutant background.

Author response: We agree with the reviewer that ASH calcium signals could arise from other neurons and have modified the text suitably. However, our genetic rescue experiments show that TYRA-2 expression in ASH neuron and also mis-expressing it in the AWA and ADL neurons is sufficient for driving both calcium change and behavior. These findings suggest that synaptic or peptidergic transmission may not be required for ASH sensation of osas#9.

Reviewer Comment-6: Page 6, graph 3, first sentence, "osas-9" should be "osas#9".

Author response: We have corrected this typo.

Reviewers' comments:

Reviewer #1 (Remarks to the Author):

The authors have addressed my previous concerns, and I support publication of this interesting manuscript.

Reviewer #2 (Remarks to the Author):

The revised version of Chute et al. addresses my major concerns. The authors have demonstrated that TYRA-2 is required for pheromone-sensing by *C. elegans* using diverse approaches. Their data strongly support a model in which TYRA-2 is the pheromone receptor. The experiments are rigorous and accompanied by appropriate statistical analysis. The implications of the authors' discovery are interesting and discussed in a manner that is frank and accessible.

I have only a few additional comments:

1. Page 11 line 15 and page 12 line 3. The authors refer to changes in calcium as 'depolarization' and 'hyperpolarization.' The observed changes in calcium might reflect changes in membrane potential, but there are other explanations for the data. They should refer to changes in calcium as such and not assume that membrane potential is changing.
2. Page 9 line 15. 'cocentrations' should be 'concentrations.'
3. Page 10 line 9. The authors cite a personal communication that suggests that the experiment they describe has been previously done by others. Is this the case? Please clarify what was communicated.
4. Page 16 line 14. In the methods, please clarify what is meant by 'Integrated mutant strains.' Are these strains with integrated transgenes? Or do the authors mean that mutants and controls were mixed together and handled in a batch?

Reviewer #3 (Remarks to the Author):

The authors have addressed some of my comments, but left some other questions untouched. For example, is *osas#9* a ligand for TYRA-2? It is common in the *C. elegans* field to express GPCRs in cell lines to test if they are receptors for a ligand. The authors did not even make an attempt to test this.

The authors also refused to test *unc-13* and *unc-31* mutants. This is a key experiment to determine if ASH neurons directly sense *osas#9*, which cannot be replaced by their rescuing data, as the rescuing data only shows that TYRA-2 functions in ASH. This is a simple experiment, so I am not sure why the authors did not want to perform it.

Reviewers' comments:

Reviewer #1 (Remarks to the Author):

The authors have addressed my previous concerns, and I support publication of this interesting manuscript.

Author response: We are glad that the revised version has addressed the reviewer's concerns and thank the reviewer for supporting publication of our manuscript.

Reviewer #2 (Remarks to the Author):

The revised version of Chute et al. addresses my major concerns. The authors have demonstrated that TYRA-2 is required for pheromone-sensing by C. elegans using diverse approaches. Their data strongly support a model in which TYRA-2 is the pheromone receptor. The experiments are rigorous and accompanied by appropriate statistical analysis. The implications of the authors' discovery are interesting and discussed in a manner that is frank and accessible.

Author response: We thank the reviewer for their positive comments on our manuscript.

I have only a few additional comments:

Reviewer Comment-1: Page 11 line 15 and page 12 line 3. The authors refer to changes in calcium as 'depolarization' and 'hyperpolarization.' The observed changes in calcium might reflect changes in membrane potential, but there are other explanations for the data. They should refer to changes in calcium as such and not assume that membrane potential is changing.

Author response: We agree with the reviewer that changes in calcium does not imply changes in membrane potential. We have now edited the two sections to suggest that osas#9 stimulation results in changes in calcium levels in the different neurons tested, rather than changes in membrane potential.

Reviewer Comment-2: Page 9 line 15. 'cocentrations' should be 'concentrations.'

Author response: We have corrected the typo in the revised manuscript.

Reviewer Comment-3: Page 10 line 9. The authors cite a personal communication that suggests that the experiment they describe has been previously done by others. Is this the case? Please clarify what was communicated.

Author response: Previous studies on different G protein-coupled receptors suggest that proper trafficking of the translational construct allows for better ciliary localization of the protein. The Sengupta lab in particular, has observed trafficking problems of GPCR proteins, when the concentration of the transgene is very high. They titrate the concentration of the protein to ensure the proper transport of the receptor to the cilia.

Hence, based on their suggestion we injected a very low concentration of the construct to characterize the expression of TYRA-2 localization.

Reviewer Comment-4: Page 16 line 14. In the methods, please clarify what is meant by 'Integrated mutant strains.' Are these strains with integrated transgenes? Or do the authors mean that mutants and controls were mixed together and handled in a batch?

Author response: We agree that usage of the phrase 'integrated mutant strains' is confusing. In cases, where an extrachromosomal transgene containing line was crossed into a genetic mutant background, we used the phrase 'integrated mutant strain'. We have now edited the section. and changed to "Strains containing transgenes in different genetic backgrounds and controls are prepared using common M9 buffer to wash and transfer a plate of animals to a microcentrifuge tube where the organisms are allowed to settle".

Reviewer #3 (Remarks to the Author):

Reviewer Comment-1: The authors have addressed some of my comments but left some other questions untouched. For example, is osas#9 a ligand for TYRA-2? It is common in the C. elegans field to express GPCRs in cell lines to test if they are receptors for a ligand. The authors did not even make an attempt to test this.

Author response: Our manuscript reports *co-option of a canonical neurotransmitter receptor and its canonical ligand, octopamine, for a signaling cascade that communicates starvation across a population*. We do not claim that osas#9 directly binds to TYRA-2, rather, we *explicitly* discuss the possibility that TYRA-2 is part of a GPCR heterodimer/oligomer complex that senses osas#9 (which is the case for other small molecule receptors in *C. elegans*, see e.g. Park, O'Doherty et al., PNAS 2012). Hence, we do not agree that demonstration of direct binding of osas#9 to TYRA-2 in *Xenopus* or cell culture is necessary to support the conclusions presented in our manuscript.

As we mentioned in our previous rebuttal, we have been collaborating with the Forrester Lab (U Ontario), to test for direct binding of osas#9 to TYRA-2 in *Xenopus* oocytes. However, we've only been able to achieve very low levels of TYRA-2 expression in this system, and we have not been able to detect any above-background currents in response to either the canonical ligand, tyramine, or osas#9. It is possible that we'd need to co-express a specific *C. elegans* G protein or a dimerization GPCR, or that human cell lines are more suitable. We believe that a full reconstitution of the osas#9 sensation machinery that TYRA-2 is part of is beyond the scope of the current manuscript, but rather an interesting and exciting avenue for future research.

Reviewer Comment-2: The authors also refused to test unc-13 and unc-31 mutants. This is a key experiment to determine if ASH neurons directly sense osas#9, which cannot be replaced by their rescuing data, as the rescuing data only shows that TYRA-2 functions in ASH. This is a simple experiment, so I am not sure why the authors did not want to perform it.

Author response: We thank the reviewer for suggesting this experiment. We have generated imaging lines in both genetic backgrounds; *unc-13* (synaptic) and *unc-31* (peptidergic). Our results clearly show that osas#9 stimulation elicits calcium transients in ASH neurons in both *unc-13* (synaptic) and *unc-31* (peptidergic) mutant worms. ASH connectivity is strongly perturbed in both these backgrounds, which can affect the magnitude of calcium transients due to lack of synaptic or peptidergic feedback respectively. In fact, we observe a weaker response in these animals than in wild type background, suggesting that a distributed circuit modulates osas#9 dynamics, as has been previously observed for other chemical stimuli in *C. elegans* (Guo et al., *Nat. Comm.* 2015, Leinwald et al., *eLife* 2015).

Figure 1: ASH sensory neuron responses upon stimulation with 1µM osas#9 in synaptic (*unc-13*) and peptidergic (*unc-31*) mutant backgrounds. Both mutants show significant increases in calcium levels upon osas#9 exposure, though the fluorescence does not decrease upon stimulus removal. ** p < 0.01, Student's t-test.

REVIEWERS' COMMENTS:

Reviewer #3 (Remarks to the Author):

The authors have finally addressed my remaining comments. The 10-fold reduction in osas#9-activated ASH calcium response in unc-13 and unc-31 is a concern, as it shows that most osas#9-evoked ASH responses are probably triggered by other neurons. Nevertheless, ASH still retains some small responses in unc-13 and unc-31 mutants. This is an important piece of information. Overall, the authors did a nice job and this is an interesting paper. I am happy to support its publication in Nat Comms. Congratulations!

REVIEWERS' COMMENTS:

Reviewer #3 (Remarks to the Author):

Reviewer Comment: The authors have finally addressed my remaining comments. The 10-fold reduction in osas#9-activated ASH calcium response in unc-13 and unc-31 is a concern, as it shows that most osas#9-evoked ASH responses are probably triggered by other neurons. Nevertheless, ASH still retains some small responses in unc-13 and unc-31 mutants. This is an important piece of information. Overall, the authors did a nice job and this is an interesting paper. I am happy to support its publication in Nat Comms. Congratulations!

Author response: We thank the reviewer for their positive comments and support for publication of the revised version.